# CIRCUIT COMPONENT REUSE ACROSS TASKS IN TRANSFORMER LANGUAGE MODELS

**Jack Merullo**
Department of Computer Science
Brown University
jack_merullo@brown.edu

**Carsten Eickhoff**
School of Medicine
University of Tübingen
carsten.eickhoff@uni-tuebingen.de

**Ellie Pavlick**
Department of Computer Science
Brown University
ellie_pavlick@brown.edu

## ABSTRACT

Recent work in mechanistic interpretability has shown that behaviors in language models can be successfully reverse-engineered through circuit analysis. A common criticism, however, is that each circuit is task-specific, and thus such analysis cannot contribute to understanding the models at a higher level. In this work, we present evidence that insights (both low-level findings about specific heads and higher-level findings about general algorithms) can indeed generalize across tasks. Specifically, we study the circuit discovered in Wang et al. (2022) for the Indirect Object Identification (IOI) task and 1.) show that it reproduces on a larger GPT2 model, and 2.) that it is mostly reused to solve a seemingly different task: Colored Objects (Ippolito & Callison-Burch, 2023). We provide evidence that the process underlying both tasks is functionally very similar, and contains about a 78% overlap in in-circuit attention heads. We further present a proof-of-concept intervention experiment, in which we adjust four attention heads in middle layers in order to 'repair' the Colored Objects circuit and make it behave like the IOI circuit. In doing so, we boost accuracy from 49.6% to 93.7% on the Colored Objects task and explain most sources of error. The intervention affects downstream attention heads in specific ways predicted by their interactions in the IOI circuit, indicating that this subcircuit behavior is invariant to the different task inputs. Overall, our results provide evidence that it may yet be possible to explain large language models' behavior in terms of a relatively small number of interpretable task-general algorithmic building blocks and computational components.[1]

## 1 INTRODUCTION

Neural networks are powerful but their internal workings are infamously opaque. Recent work in mechanistic interpretability explains specific processes within language models (LMs) using *circuit analysis* (Wang et al., 2022; Hanna et al., 2023; Lieberum et al., 2023) which reverse-engineers small subnetworks that explain some behavior on a dataset. Through causal interventions, these studies are able to attribute interpretable roles for specific model components involved in predicting the correct answer for the task. A major criticism of this work is that while these studies explain the exact task being studied very well, it is unclear how they help our understanding of model behavior beyond that domain. While current evidence suggests that there is at least some reuse of highly general components like induction heads (Olsson et al., 2022), it has yet to be shown whether larger structures recovered in circuit analysis can be repurposed for different tasks. In the most pessimistic case, every different task is handled idiomatically by the model. If this were true, having a circuit for every task would leave us no better off from an interpretability standpoint than having the full model itself.

---

[1]Code available at: https://github.com/jmerullo/circuit_reuse

To better understand this, we study whether LMs reuse model components across different tasks to accomplish the same general behaviors, and whether these compose together in circuits in predictable ways. We study two tasks that have no obvious linguistic overlap but that we hypothesize may require similar processes to solve. One is the Indirect Object Identification (IOI) task, for which Wang et al. (2022) discover a circuit in GPT2-Small (Radford et al.). The other is the Colored Objects task (Figure 1), which is a variation on an in-context learning task from BIG-Bench (Ippolito & Callison-Burch, 2023). To solve the Colored Objects task, a model must copy a token from a list of possible options in context. Based on the interpretation of the IOI circuit, which performs a behavior along those lines, a simple way to test the idea of neural reuse would be to see if these two tasks use the same circuit. We perform path patching (Wang et al., 2022; Goldowsky-Dill et al., 2023) on both of these tasks on GPT2-Medium (Sections 3 and 4) and compare their circuits. Our causal interventions show that the Colored Objects circuit uses largely the same process, and around 78% of the 'in-circuit' attention heads are shared between the two tasks[2]. Such a high degree of overlap supports the idea of general-purpose reuse. We use this insight to design 'ideal' interventions on the model that both improve the Colored Objects performance from 49.6% to 93.7% and provide evidence that at least part of the original IOI circuit appears to be part of a more generic circuit for controlling selection among competing alternatives in context.

Our contributions are summarized below:

1. In Section 3 we reproduce the IOI circuit on GPT2-Medium, showing that this circuit has been learned by more than one model. We also expand on the understanding of the role and functionality of inhibition and negative mover heads in models.

2. In Section 4 we perform a circuit analysis on the Colored Objects task, and show that the GPT2 breaks it down into largely the same principal steps as the IOI task, using approximately 78% of the same most-important attention heads to do so.

3. In Section 5, by intervening on the inactive parts of the Colored Objects circuit to make it act more like the IOI circuit, we increase task accuracy from 49.6% to 93.7%. More importantly, we empirically show that these interventions have the downstream effect that would be predicted by the interactions in the IOI circuit, showing that the inhibition-mover head subcircuit is a structure in the model that is robust across changes in the input task.

## 2 EXPERIMENTAL SETUP

In this work, we show that the circuit which solves Indirect Object Identification (IOI) task from Wang et al. (2022) uses most of the same components as the circuit for a seemingly different task. We use path patching (Wang et al., 2022; Goldowsky-Dill et al., 2023), attention pattern analysis, and logit attribution to reverse engineer and explain the important model components necessary to predict the correct answer. The two tasks we use are described below. We briefly describe our methods below (§2.2), and in greater depth in Appendices A and B.

### 2.1 TASKS

**Indirect Object Identification:** We use the IOI task from Wang et al. (2022), which requires a model to predict the name of the indirect object in a sentence. The following example is representative of the 1000 examples we use in our dataset: "Then, Matthew[IO] and Robert[S1] had a lot of fun at the school. Robert[S2] gave a ring to" where the LM is expected to predict "Matthew". The models we analyze perform well on this task, preferring the IO token to the Subject token in the logits 100% of the time.

**Colored Objects:** The Colored Objects task requires the model to generate the color of an object that was previously described in context among other other objects. An example is shown in Figure 1. We modify the Reasoning about Colored Objects task from the BIG-Bench dataset (Ippolito & Callison-Burch, 2023)[3] to make it slightly simpler and always tokenize to the same length. There

---

[2]Quantifying overlap is a difficult problem that we do not have an exact solution for. We discuss this in Appendix I.3.

[3]https://github.com/google/BIG-bench/tree/main/bigbench/benchmark_tasks/reasoning_about_colored_objects

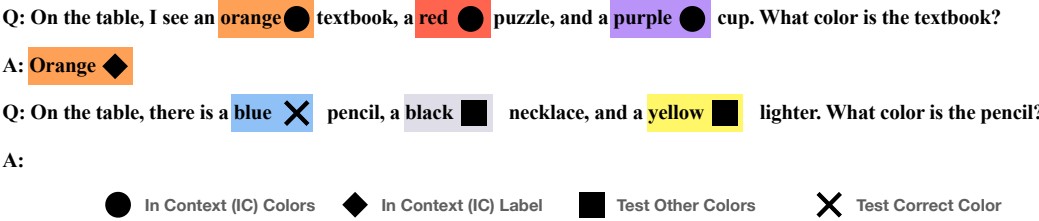

Figure 1: An example from the the modified Colored Objects task. All inputs are one shot, where the first example is the In Context (IC) example, and the second is the Test example. The goal of the task is to predict the correct color (denoted by the 'X') and ignore the other color options (in particular, the other test example colors, denoted by a square).

are 17 total object classes and only one object of any type appear within an example. There are eight possible colors (orange, red, purple, blue, black, yellow, brown, green). No objects have the same color within any example. We generate a dataset of 1000 examples and for path patching, generate 1000 variations of these differing only in which of the three objects is asked about. Further details can be found in Appendix H. To encourage the model to only ever predict a color token as the next token, we always provide one in-context example. We find that this is enough signal for the model to predict a color as the next token: 100% of the time, the model will predict one of the three colors in the test example as the next word (Figure 1 for reference). However, GPT2-Medium does not perform consistently well, only achieving 49.6% accuracy. In Section 5, we investigate this poor performance and design interventions which offer further insights into the circuit components in play in this and the IOI task.

## 2.2 PATH PATCHING

Path patching Wang et al. (2022); Goldowsky-Dill et al. (2023) is a causal intervention method for reverse-engineering circuits in networks. It involves replacing the activations from model component(s) (e.g., attention heads) on input $x_{original}$ with those of another input $x_{new}$ ("patching"). This approach allows us to find attention heads that have a direct causal effect on the model's predictions. To summarize, we start by finding heads that have the largest direct effect on the model logits. That is, an attention head $h$ has a large direct effect if swapping the value of $h$ when processing $x_{original}$ with the value that $h$ takes when processing $x_{new}$ causes a large drop in logit difference between the answers of the two inputs, $y_{original} - y_{new}$. From there, we find the heads that largely impact these heads, and work backwards until we have explained all the behaviors of interest to the present study. As in the original IOI paper, we only patch paths between attention heads, allowing the MLPs to be recomputed. A more detailed explanation of path patching and our approach can be found in Appendix A.

## 3 THE INDIRECT OBJECT IDENTIFICATION (IOI) CIRCUIT

Because of the poor performance of GPT2-Small on the Colored Objects task, we use the larger GPT2-Medium model. This means we must first reproduce the IOI results from Wang et al. (2022) on the larger model. We find that we are able to replicate the circuit described in the original paper on the new model; all of the components described in that work are active for the larger model as well, with minor differences allowing for better performance described below. More details on our results are found in Appendix C. We will use the following as our running example from the IOI task to explain the circuit: **"Then, Matthew[IO] and Robert[S1] had a lot of fun at the school. Robert[S2] gave a ring to (Matthew)"**. At a high level, the IOI circuit implements the following basic algorithm: 1) Duplicate names (S1 and S2) are detected by **Duplicate Token/Induction Heads**; 2) The detection of the duplicates informs the **Inhibition Heads** to write an inhibitory signal about these tokens/positions into the residual stream; 3) This signal instructs the query vectors of the **Mover Heads**, telling them to *not* attend to these tokens/positions. Mover heads write into the residual stream in the direction of the token that they attend to (in simple terms, tell the model to predict whatever they attend to). The mover heads in IOI attend to names, and because of the inhi-

bition signal on S1 and S2, the remaining name (IO) receives more attention instead. As a result, this token is copied into the residual stream (Elhage et al., 2021), causing the model to predict that token. The role of these components in IOI were described first in Wang et al. (2022). We show that part of the circuit described here boils down to a more generic algorithm for copying from a list of potential options, rather than strictly indirect object identification.

**Negative Mover Heads** Our reproduced IOI circuit involves one active *negative* mover head (19.1). This head does the opposite of the other mover heads: whatever it attends to, it writes to the logits in the opposite direction. Wang et al. (2022) find that the negative mover heads in GPT2-Small attend to all names and hypothesize that they hedge the prediction to avoid high loss. In contrast, we find that in GPT2-Medium, this head attends only to the S2 token and demotes its likelihood as the next prediction (Appendix C). In fact, it is the *most* important head contributing to the logit difference in path patching. Our results suggest that the negative mover head in GPT2-Medium is perhaps capable of more sophisticated behavior, by directly demoting the subject token without interfering with the role of the other mover heads.

## 4 THE COLORED OBJECTS CIRCUIT

Given the above understanding of the IOI circuit, we now ask whether any of the discovered circuitry is repurposed in the context of a different task, namely, the Colored Objects task. Again, for space, the full analysis is available in Appendix D. We will use the following as our running example from Colored Objects: **"Q: On the table, I see an orange textbook, a red puzzle, and a purple cup. What color is the textbook? A: Orange Q: On the table, there is a blue[$col_1$] pencil[$obj_1$], a black necklace, and a yellow lighter. What color is the pencil[$obj_2$]? A: (Blue)"** On one level, these tasks are completely different (they depend on different syntactic and semantic structure), but on another level, they are similar: both require the model to search the previous context for a matching token, and copy it into the next token prediction. We are thus interested in whether the LM views these tasks as related at the algorithmic or implementation level.

Using the methods described above (§2.2), we find that the principle 'algorithm' used by GPT2-Medium to solve this task is as follows. Details of the evidence supporting these steps is found in the following subsections

1. **Duplicate/Induction Heads** (§4.3) detect the duplication of the $obj_1$ token at the $obj_2$ position.

2. **Content Gatherer Heads** (§4.2) attend from the [end] position (the ':' token) to the contents of the question, writing information about the token 'color' and $obj_2$ into the residual stream. These heads (ideally) provide information about which specific object is the correct answer, and thus provides a positive signal for which token to copy to the mover heads. The role of these heads is qualitatively similar to those in Lieberum et al. (2023), thus we use the same term here. This step appears to replace the inhibition step from IOI (§3).

3. After the information about the question has been collated in the final token, **Mover Heads** (§4.1) attend from the final position to the three color words and write in the direction of the one most prominently attended to. We do not find that there any *negative* mover heads contributing significantly to the logits.

The above process is remarkably similar algorithmically to that used for IOI: the detection of duplication informs the mover heads either where or where *not* to look. Figure 3 visualizes this overlap and the relative importance of each head, which shows a very large overlap between the heads performing this function; thresholding at the 2% most important heads for each circuit, we find that 25/32, or 78% of the circuit is shared. This is a difference of swapping out three inhibition heads and a negative mover head for three content gatherer heads. It is worth noting that such overlap is not trivial–i.e., it is not as though any two circuits will contain the same components or even overlapping algorithmic steps. See, for example, the circuit described for the greater than task in Hanna et al. (2023), which looks fundamentally different[4]. In the following sections, we will describe the

---

[4] compared to IOI on GPT2-Small there is essentially no overlap in specific behaviors and heads

evidence for the roles we prescribe to components in circuit, before returning to evaluating the differences to the IOI circuit. For consistency with the path patching process, we decribe components working backward from the end of the algorithm to the beginning.

## 4.1 MOVER HEADS

Path patching to the logits identifies heads at 15.14 (Layer 15, Head 14), 16.15, 17.4, 18.5, 19.15, among others as mover heads. We find that many of the heads in the later third of GPT2-Medium are contributing to copying. This is one of the most obvious points of overlap we find between these two circuits. In fact, we find that most of the exact same heads (e.g., the most important 15.14, 16.15) which were responsible for the final copy step in IOI are also responsible for the final copy step in Colored Objects. While the relative importance differs (e.g., 18.5 is the much more important for IOI and 15.14 is the most important for Colored Objects), there is nonetheless a remarkable amount of component level overlap. This is the first piece of evidence we provide that model components are generic and are reused across different tasks. We repeat the same analysis done on the IOI mover heads in Appendix D.1 where we show that these heads preferentially attend to color tokens in the test example, promote the prediction of those words by writing in their embedding directions, and are the same as those used by the IOI circuit.

## 4.2 CONTENT GATHERER HEADS

Path patching to the query vectors of the mover heads reveals that heads 11.6, 11.7, and 12.15 (and to a lesser extent, 13.3) are performing the role of content gatherer heads. This is a major difference from the IOI circuit. Namely, the Colored Object circuit relies on content gatherer (CG), rather than inhibition heads for influencing the query vectors of the mover heads. While inhibition heads tell the mover heads which tokens to ignore, the content gatherer heads tell the mover heads which tokens/positions to focus on. Collectively, these heads attend primarily to the $obj_2$ token and the token 'color' in the question (Figure 2).

To confirm that they are playing this role, we run an intervention as follows: at the [end] position, we block attention to the words in the question part of the test prompt about which we hypothesize the CGs are passing information. In particular, if these heads account for all of the signal that tells the model which object is being asked about, then the mover heads should randomly copy between the three color words in the test example. By doing so, accuracy is reduced to 35%, i.e., the model is selecting the correct color from among the three test example colors roughly at random chance. For three seeds, we randomly sample three heads from these layers (without replacement) that are not predicted as playing a role in content-gathering and repeat this intervention the average accuracy for random heads is 49.9%±1.0. Further analysis on these heads is done in Appendix D.2.

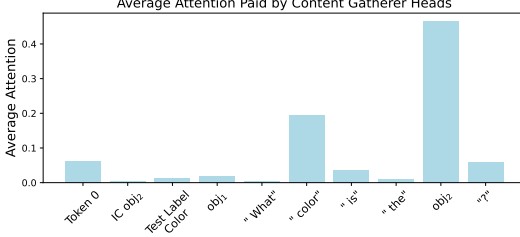

Figure 2: Average attention paid to tokens in the input by content gatherer heads. Most attention is paid to the $obj_2$ token and the word ' color'.

## 4.3 DUPLICATE/INDUCTION HEADS

The $obj_2$ token is the most heavily attended to and important token for passing signal to the [end] position via CG heads (see previous section). We find that path patching to the value vectors of the CG heads has the biggest impact on the logit difference at this position, indicating that they are most impacted by duplicate token heads. As inhibition heads in IOI are activated by the detection of a duplicate mention of a name, signalling for the inhibition of attention to that token, this same duplicate token signal influences the values of the content gatherer heads. By path patching to the value vectors of the content gatherer heads at the $obj_2$ position, we find that induction heads like 9.3 and duplicate token heads like 6.4 attend from $obj_2$ to $obj_1$ or the $obj_1$+1 token (due to how induction heads move information, see Olsson et al. (2022); Wang et al. (2022)). This signal helps the model find where to look to find the final color token. These heads overlap significantly with those in IOI (Figure 3).

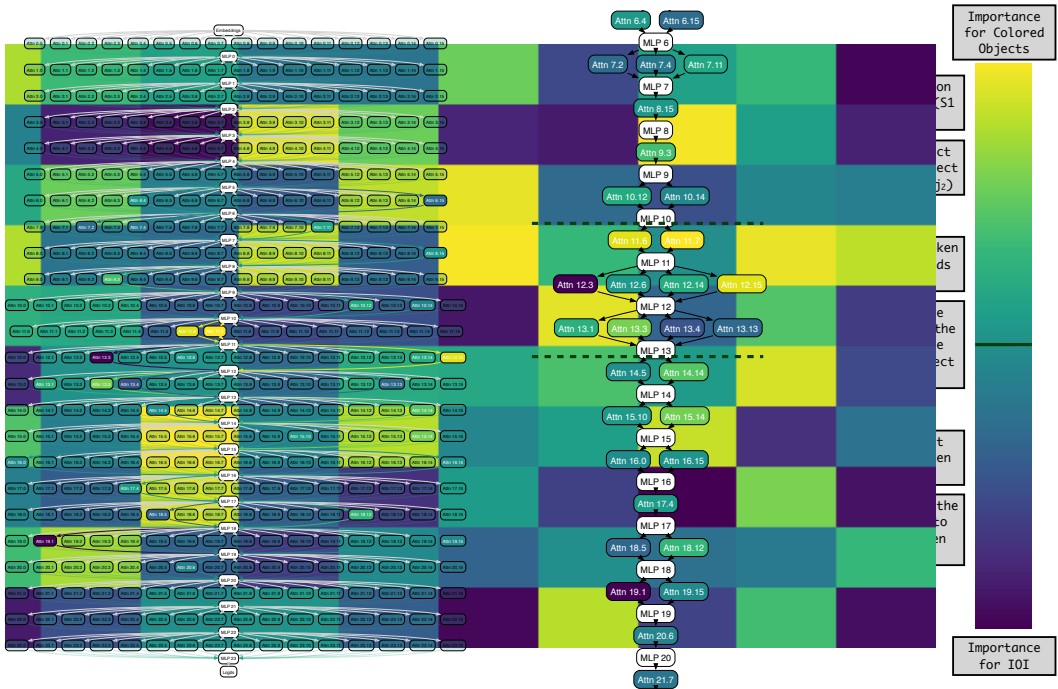

Figure 3: Left: The full graph of attention heads with the difference of the path patching importance scores between each task (normalized by task per each path patching iteration). Middle: Visualizing only the union of the top 2% most important heads (per path patching stage) for each task, colored by the difference in importance scores. Right: Explanation of each stage of processing in the circuit. Both circuits involve the same general process: detecting a duplication and using that duplication to decide which token to copy. In the Colored Objects task, the duplication is used as a "positive" signal via the content gatherer heads to tell the mover heads which token to copy, while in IOI the duplication sends a "negative" signal via the inhibition heads to tell the mover heads which tokens to ignore. These heads, and the activation of the negative mover head in IOI constitute the only major difference between the two tasks.

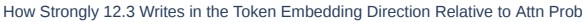

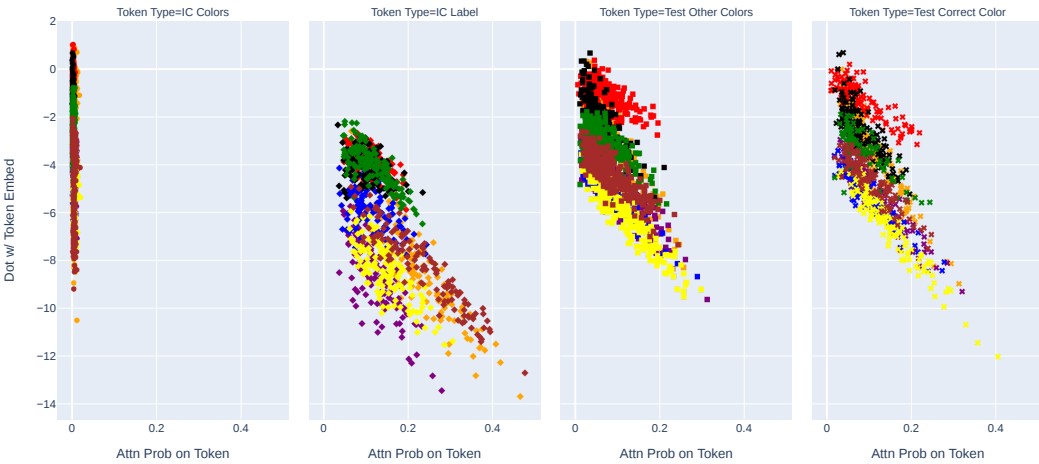

Figure 4: Analyzing one of the inhibition head's (12.3) activity on the Colored Objects task shows that it is attending strongly to test color words and the in-context label (and writing in the opposite direction in embedding space, when attention is high), although they do not affect the mover heads as they do in IOI. Scatter plots for the other two inhibition heads are shown in Appendix D.3. Colors indicate the color of the word being attended to. See Appendix B for explanations of the axes.

### 4.4 IOI Components Missing in Colored Objects

In comparing the Colored Objects circuit above with the IOI circuit, the most notable differences are the addition of content gatherer heads and the absence of the Inhibition and Negative Mover Heads. Investigating further, we find that these missing heads are in fact active in Colored Objects, but are receiving incorrect biases and promoting a noisy signal which does not help the model find the answer in the logits. We elaborate on this analysis below, and use it to motivate a proof-of-concept intervention on the Colored Objects task in Section 5.

**Inhibition Heads** Inhibition heads prevent the mover heads from attending to some token or position written into the residual stream (Wang et al. (2022); Appendix C). From patching to query vectors of the mover heads on Colored Objects, we see that the inhibition heads are actually mildly working *against* the model: One would expect patching in another activation into the heads would hurt performance, but it actually improves logit difference from around 1-3%. We explore why this is the case here. We find that this inhibition signal actually does exist for the Colored Objects task, but exhibits a noisy signal that prevents it from having a useful impact on the prediction. Figure 4 shows the attention to color tokens against the writing direction of inhibition head 12.3. We observe that the model is able to recognize reasonable places to place an inhibition signal: On the answer to the in context example (the IC label), and on the test example color tokens. However, the model spreads the inhibition signal across all of these and shows no bias to attend to the *wrong* answers as we might expect the model to do. This is notable because in the IOI task the inhibition heads depend on duplication to know what to attend to, whereas here they are attending to color tokens without those tokens having been duplicated in the input. We hypothesize that the inhibition heads would help improve performance the way they do in the IOI task if they were to get the right signal, which we explore in Section 5.

**Negative Mover Heads** On the Colored Objects task the negative mover head (19.1) does not heavily impact the logits, however it does seem to be performing negative token moving (Appendix D.4). The head places most of its attention on the very first token (anecdotally, this has been reported as indicating that it is "parked" i.e., doing nothing, see Kobayashi et al. (2020) for a related result in encoder-only models), with typically less than 5% attention to the color tokens in the test example and the in-context example's label token (the token after the previous "A:").

### 4.5 Summary of Similarities and Differences

Figure 3 summarizes the role of each head and shows the overlap over the entire model and in the top 2% of important heads (which we deterimine to be the minimal threshold for containing the in-circuit components for both tasks). Despite the differences in the task structure, the Colored Objects circuit resembles that used by the model to solve IOI. We observe that the mover and induction head components not only act consistently between the two tasks: using duplication detection to decide on which token to promote with the mover heads, but the exact heads for each function in the circuits are largely the same. We find that the inhibition heads, which are active and contributing to the circuit in IOI, are also active in Colored Objects, but receiving incorrect biases and promoting a noisy signal which does not help the mover heads find the answer downstream.

In the following sections, we show that when we intervene on the model to make to make them behave more like the IOI circuit, task accuracy increases dramatically, and the downstream components change their behavior in the expected ways (as a result the inhibition intervention in particular). We use this evidence to argue that the model learns a modular subcircuit for inhibition and copying that it is able to use across task domains.

## 5 Error Analysis on Colored Objects

As previously described, the accuracy of GPT2-Medium on the Colored Objects task is low (49.6%). Intuitively, an inhibition signal would be very helpful for the model, as 100% of the time the model predicts that the answer is one of the three colors from the test example. This indicates that the mover heads have trouble selecting the right color over the distractors. According to the analysis on the IOI task, inhibition heads are critical for solving this problem, but they do not give a useful

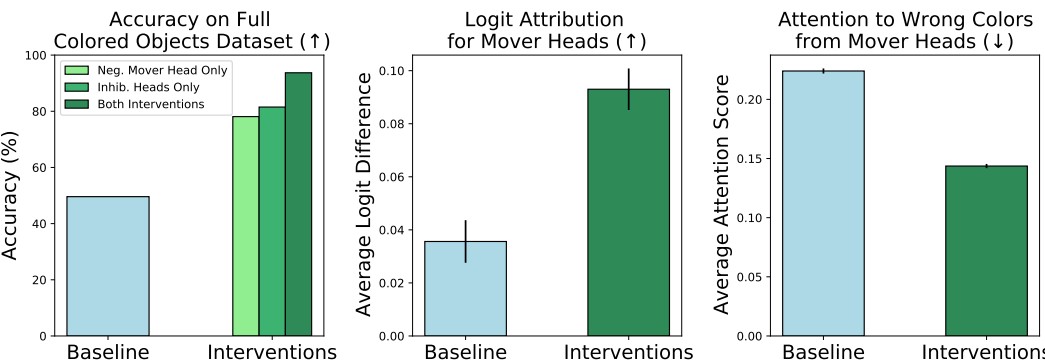

Figure 5: Intervening on the attention patterns of the inhibition heads and negative mover increase accuracy on the full dataset from 49.6% to 93.7%. Furthermore, the interventions (specifically on the inhibition heads) affect the mover heads in the ways predicted by the IOI circuit. The right two graphs show a comparison of the logit difference and the attention to wrong colors before and after the intervention; results for these two are taken only over the 496 examples GPT2-Medium originally gets right, for a fair comparison. This evidence together suggests that the inhibition-mover subcircuit is itself a manipulable structure within the model that is invariant to the highly different input domains that we used in our experiments. Error bars show standard error.

signal in Colored Objects. Likewise, the negative mover head is also not active. Given the results that other attention heads are being reused for similar purposes, it should follow that intervening on these to provide the right signal should improve the model performance and recover the missing parts of the IOI copying circuit.

In this section, we intervene on the model's forward pass to artificially activate these model components to behave as the IOI task would predict they should. To do this, we intervene on the three inhibition heads (12.3, 13.4, 13.13) and the negative mover head (19.1) we identify, forcing them to attend from the [end] position (":" token) to the incorrect color options. Consider the example in Figure 1: we would split the attention on these heads to 50% on the " black" and " yellow" tokens. We hypothesize that the model will integrate these interventions in a meaningful way that helps performance.

With the attention pattern intervention described above, we are able to improve accuracy from 49.7% to 93.7% with both interventions, to 78.1% with just the negative mover head, and 81.5% with just the inhibition head interventions (Figure 5). Importantly, the intervention introduces zero new mistakes. We find that the negative mover head does simply demote in the logits whatever it attends to. The IOI-copying behavior demands that the change to the inhibition heads is affecting the previously identified mover heads. In Appendices C.3, D, and G, we show that inhibition head 12.3 in particular does have some effect on the logits in both tasks, but not enough to explain the performance increase. To conclusively show that the IOI copying behavior is recovered, we analyze their downstream effect on the mover heads.

If the IOI circuit is involved the way we hypothesize it is for Colored Objects, then, as a result of the inhibition head intervention, we should see the attention of the mover heads to the incorrect colors decrease and the logit attribution of those heads to increase. In Figure 5, we see exactly that. The attention to the incorrect colors decreases significantly (on average -8.7%), while attention to the correct color remains high, or increases (on average 2.7%, shown in Appendix G). We also find that the logit attribution of mover heads increases on average (about 3x). Across all heads, mover heads are particularly correlated with a higher logit attribution as a result of the intervention: The Spearman correlation between the change in logit attribution of heads after the intervention (excluding all intervened heads) and their original path patching effect on the logits (i.e., mover heads) is 0.69 ($p < 0.01$). Overall, the results suggest that the intervention helps the model recover the rest of the circuit that was first observed on the IOI task and explains how the performance increases so drastically.

## 6    RELATED WORK

In transformer language models, there are obvious examples of reuse: simple components like induction heads, previous token heads, and duplicate token heads (Olsson et al., 2022). While these implement specialized functions, it has been shown that their functionality is reused to contribute in non-obvious ways to arbitrary tasks, such as in in-context learning. These contrast with specialized components that activate or change *specific* conceptual features, e.g., neurons that detect the French language (Gurnee et al., 2023), or components storing specific factual associations (Meng et al., 2022; Geva et al., 2021). More broadly, the ways attention heads are used by a model to build predictions in models has been widely studied (Voita et al., 2019; Vig, 2019) and more recently, using causal interventions, (Pearl; Vig et al., 2020; Jeoung & Diesner, 2022), path patching Wang et al. (2022); Goldowsky-Dill et al. (2023), and other circuit analysis techniques (Nanda et al., 2022; Conmy et al., 2023; Elhage et al., 2021).

## 7    CONCLUSION AND FURTHER DISCUSSION

### 7.1    CONCLUSION

A major concern with the methods of mechanistic interpretability we address in this work is the possibility that analyzing circuits in smaller models and/or on toy tasks might not lead to better understanding of neural networks in general. Our work provides a proof of concept where results from an analysis done on a small model (GPT2-Small) on a toy task (IOI) in Wang et al. (2022) allows us to make accurate predictions about a different and more complex task task on a bigger version of that model (GPT2-Medium). Aligning the IOI circuit onto GPT2-Medium revealed a simple and obvious avenue via which we could 'ideally' intervene on the model to fix model errors. Although the scale of the model tested here is still relatively small, related work has already shown that circuit analysis techniques can help us understand models in the billions of parameters range (Lieberum et al., 2023). That, combined with the insights we present here, suggests an exciting path forward for understanding, controlling, and improving even production-scale models.

### 7.2    PREDICTING MODEL BEHAVIORS

The behavior of LMs is infamously hard to predict. Current auditing work only catches 'bad' behavior after a model has already been deployed. If we know what the core algorithmic steps that LMs employ are and how they tend to decompose complex functions into those steps, we can better anticipate their behavior and predict what problems they will and won't solve well. A core promise of interpretability research is the ability to understand LMs at a high enough level that we can predict how changes will make them behave on new data. Our work shows an interesting case where we achieve a higher level understanding of a larger model behavior built on lower level analyses of a small model, though Appendix J suggests that showing this might not be trivial.

Connecting similarities in low-level analyses like circuits can lead us to understanding more abstract processes that underlie model behaviors, like those described in Holtzman et al. (2023). For example, knowledge general inhibition-mover interactions could help explain selection among multiple alternatives in a context.

### 7.3    WHY CAN'T GPT2 ACTIVATE THE INHIBITION HEADS ON ITS OWN?

What prevents the inhibition heads from consistently attending to the right colors in the middle layers? We speculate that there is a bottleneck in the model, where the signal propagated by the content gatherer heads, telling the model which object is being asked about, does not become fully formed before the inhibition heads are called. Such a bottleneck in a small model like GPT2-Medium could partly explain why scale tends to help models solve more complex tasks: because either the content gathering signal is fully formed before the inhibition heads are called, or because heads that act as inhibition heads are implemented redundantly in deeper networks (Tamkin et al., 2020; Merullo et al., 2023). This could also motivate work on training recurrent models, in which, for example, inhibition heads could activate on a second pass if they could not get a strong signal in the first. Such a case has been argued for in vision applications (Linsley et al., 2018).

## 8 ACKNOWLEDGMENTS

We would like to thank Aaron Traylor, Catherine Chen, Michael Lepori, Etha Hua, and members of the Google Deepmind mechanistic interpretability team for feedback on this work.

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

# A  EXPLANATION OF PATH PATCHING

Which components in a model affect the value of its final prediction? How are *those* components affected by earlier parts of the model? Path patching is a causal intervention method introduced in Wang et al. (2022); Goldowsky-Dill et al. (2023) that builds on causal mediation analysis (Pearl) which has previously been used to study causal connections in neural networks (Vig et al., 2020). We provide a summary that covers how we use it in our work, see the previous literature for a more general purpose and detailed description of the process. The premise of the technique is to run a forward pass on two examples: A and B, between which the next token prediction should be different, and cache every activation from the model for both (e.g., the attention layer outputs, MLP outputs, etc.). Take for example these two inputs from the IOI dataset:

- A: Then, Matthew and Robert had a lot of fun at the school. Robert gave a ring to
- B: Then, Matthew and Robert had a lot of fun at the school. Matthew gave a ring to

These two examples are minimally different, and we would like to learn how the model computes which name should be predicted (i.e., identifying the indirect object token, Matthew or Robert). To find the circuit the model uses, we intervene on the forward pass of one of the inputs, changing individual activations to see how they affect the downstream prediction. We will first describe activation patching, which is the basis for path patching (and simpler to understand) as a preliminary for then describing path patching in more detail.

## A.1  ACTIVATION PATCHING

Let $h_a$ be some attention head output vector for the head $h$ in the cache of activations from input A. If we believe the output computed by $h$ is very important for predicting the final answer, we can run a forward pass on B and replace the output vector $h_b$ with $h_a$ and continue the forward pass as before. We can then check the logits and see if using $h_a$ had the effect of promoting the answer to A. But by doing this, it is unclear if $h_a$ had the effect of directly influencing the logits, or influenced the logits through the effect it had on some other model component downstream of it. To separate out the direct effect, we use path patching

## A.2  PATH PATCHING

In path patching, we would like to isolate the effect of patching in some value $h_a$ on some downstream component(s) like the logits, or even other attention heads. If the next token prediction vector is represented as $x$, the attention and MLP layers update $x$ by adding into it, e.g., $x = x + \text{MLP}(x)$. If we change $h_a$, $x$ will obviously be changed as well. Since downstream components use $x$ as input, if we affect the value of $x$, the later components will be affected by that change and thus introduce indirect causal effects on the prediction. To avoid this, we have to patch in every activation from the cache for B's activations if that component would be affected by the change to $x$ earlier on in the forward pass. If *every* subsequent component is unaffected by the change from $h_a$, then only the final logits will be affected by that patch. We can selectively allow the change to $x$ via $h_a$ to affect some downstream components other than the logits (like other attention heads) to get $h_a$'s direct effect on those. The general path patching algorithm requires four forward passes as follows:

- Forward pass 1: Cache all of the activations for input A
- Forward pass 2: Cache all of the activations for input B
- Forward pass 3: Select some model component(s) that you want to patch from A to B, call this the sender set $s$. Select some component(s) for which you want to quantify how changing $s$ affects them ("How does changing $s$ affect $r$?"). Call it the receiver set $r$. Run the forward pass on B, patching in $s$ with the activations from A. Patch in all downstream components in B's cache, except for those in $r$. Components in $r$ get recomputed and now account for the change made by patching $s$. Let these recomputed values be $r'$
- Forward pass 4: Run the model on B again, this time activation patching in the values for $r'$. Measure the difference in logits between the answers for A and B, this tells you how changing the path from $s$ to $r$ affects the logits.

See Wang et al. (2022) and Goldowsky-Dill et al. (2023) for more details.

In practice, we start by path patching one attention head at a time to the logits. After selecting the heads that affect the logits the most, we repeat the process, but this time path patching each head one at a time to see how they affect this selected subset of heads, etc. We follow the path patching procedure from Wang et al. (2022) closely on the larger GPT2-Medium model, meaning we allow the MLP layers to be recomputed for every pass.

## B  GLOSSARY OF TERMS

Throughout the paper, we use terms that are highly specific to this line of work. Here we take more space to provide definitions of such terms.

1. **Circuit** Although specifics differ depending on the context of its use, a circuit typically refers to the path through the components (attention heads, MLPs) of a model that compute some behavior. If we think of all of the connections through a neural network as edges in a graph, a circuit is the path through those edges that allows the model to solve some task. Olah et al. (2020) define a circuit in much more detail, although the definition used here (and more generally in other work) is a bit looser.

2. **"residual stream" / "write in the residual stream"** Transformers consist of attention and MLP blocks (ignoring LayerNorms). The outputs of these blocks are added to the inputs of the blocks through the *residual connection*. Since the next word prediction starts as the embedding for the current word, all updates to it are made through residual connections. Therefore, a useful abstraction for how information moves is through a *residual stream* (Elhage et al., 2021) where information is added to the stream by attention/MLP layers. Therefore writing to the residual stream is done by adding a vector to it. This perspective helps in understanding how path patching is implemented.

3. **[end]** refers to the last token in a sequence. For example, the " to" token in the IOI examples.

4. **"patching to the X vectors"** refers to path patching the outputs of earlier heads from one input text into the forward pass of a minimally different input text such that it affects the X vectors of some later head, where X is either the keys, queries, or values of the later head. In other words, we only allow the output of some earlier head to affect the queries/keys/values of the later head.

5. **Dot w/ Token Embed** (E.g., Figure 4). This measurement is short for "dot product with the embedding vector of a given token". In Figures 4 and 11, for example this refers to taking the dot product between the output of some attention head and the vector in the unembedding matrix for some token $t$ of interest. This tells us how much a specific head writes in the direction of a certain token, thus promoting it in the output space. See Wang et al. (2022) for more details.

6. **Attn Prob on Token** (E.g., Figure 4). This is the attention probability mass assigned by an attention head to a token in the sequence.

## C  PATH PATCHING RESULTS ON IOI

In this section we describe the process we use to reverse-engineer the IOI task on GPT2-Medium. We follow the path patching process from Wang et al. (2022) as closely as possible and are able to replicate the same circuit. When choices are made to e.g., path patch to the queries of the mover heads, it is implied that path patching to the values or keys of these

### C.1  MOVER HEADS

First, by path patching attention heads to the logits at the [end] position.

We show an example of a mover heads in Figure 7, which shows that the dot product between the token embedding for the IO token being attended to and the output of the mover head is high across the examples and correlates strongly with the attention paid to those tokens. That is: these heads

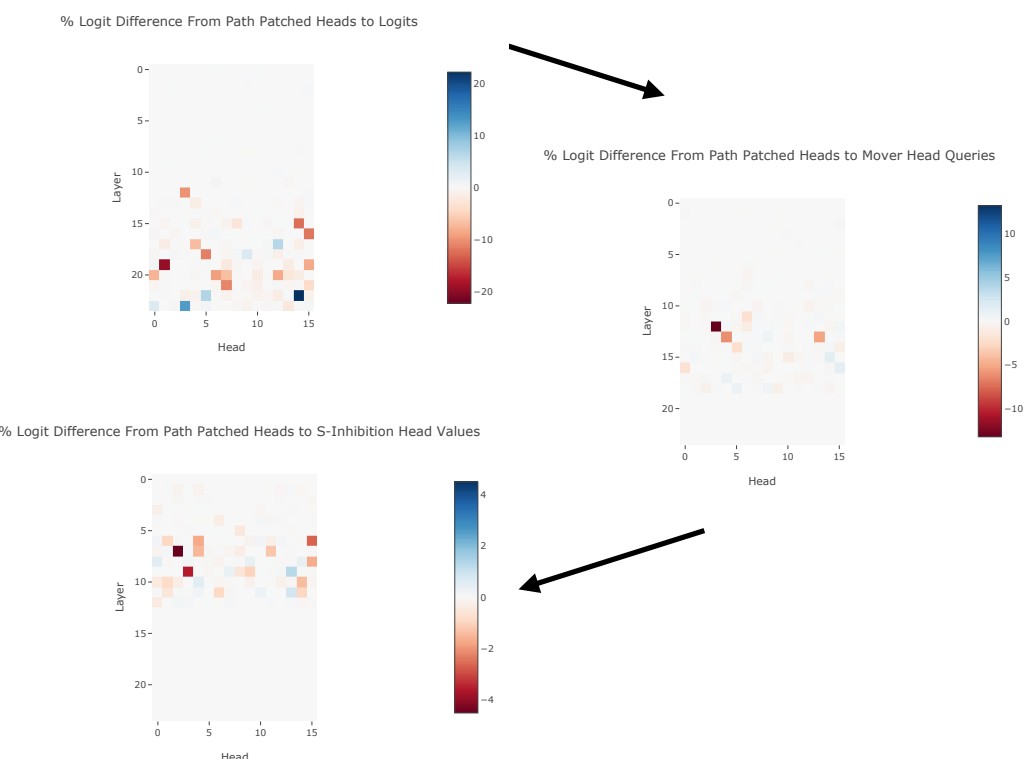

Figure 6: Full path patching results for the IOI task. Arrows show the three path patching iterations. Color bars show percent logit difference. The lower the number, the more important the head is, as patching it causes the logit difference to drop (i.e., harms performance). The heads that we find most directly affect the logits are the mover heads. Path patching to the mover head query vectors (to see how these heads 'decide' where to attend) reveals the inhibition heads (12.3, 13.4, 13.14). Path patching to the value vectors of the inhibition heads (to see what the inhibition heads write into the residual stream), we find the induction and duplicate token heads, which are sensitive to the duplication of the subject token.

copy whatever they attend to, and in this setting, they attend to the IO name in the test examples. The attention paid to and the corresponding dot products for the subject tokens is also shown, which is much lower, indicating that these are not being copied by these heads.

## C.2 The Negative Mover Head

In the main paper, we show that the negative mover head in GPT2-Medium is performing a helpful behavior: demoting the subject token in the logits without interfering with the positive mover heads. As shown in the original IOI analysis, the negative mover heads in GPT2-Small demote the IO token, not the subject token, thus hurting performance. We path patch to the query vectors of 19.1 in the IOI task to see how it chooses to attend to the subject token instead. Since it has a different attention pattern than the other mover heads it would be surprising if it relied on the same signals to decide where to attend. Figure 8 shows these results. We find that a mostly disjoint set of attention heads informs the negative mover head queries than the other mover heads, namely 16.0, 18.2, and 18.9. 16.0 and 18.2 do not appear to display any obvious pattern in what they attend to, but 18.9 seems to tend to attend to test color option tokens.

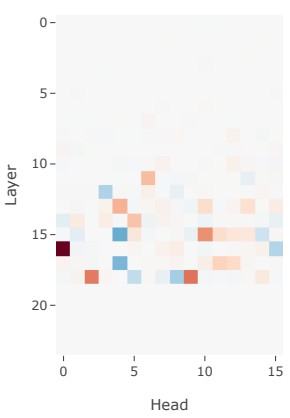

### C.3 Inhibition Heads

Following Wang et al. (2022), we path patch to the query vectors (which control what the mover heads attend to) of the mover heads with the highest logit difference (14.14,15.14,16.15,17.4,19.15). Because there are so many mover heads, we test different subsets, but find it does not make a noticeable difference. The heads that most affect these query values are the inhibition heads 12.3, 13.4, and 13.14.

Figure 8: Path patching to the queries of the negative mover head 19.1. Color bar shows percent logit difference as a result of the patching. This head relies on a different set of heads than the positive mover heads.

## D Path Patching Results on Colored Objects

We repeat the path patching process on the Colored Objects task for GPT2-Medium. We are interested in answering how the model uses the information in the prompt to predict the right color over the other possible options. As previously described, path patching requires

### D.1 Mover Heads

The first step is to path patch from each attention head to the logits at the end position. In this task this is equivalent to measuring how important each attention head is for predicting the color answer from the ":" token. Like in the IOI task, we find that there is a large set of mover heads that contribute almost all of the logit difference. Two heads in particular: 15.14 and 16.15 contribute 28 and 13 % of the logit difference respectively, making them by far the largest contributors to the logit difference. Like in IOI, 17.4, 18.5, and 19.15 also contribute significantly. The negative mover head 19.1 only contributes abut 1% logit difference, which differs from its role in IOI.

We show examples of mover heads on the Colored Objects task in Figure 11, which shows that the dot product between the token embedding for color words being attended to and the output of the mover head is high, and correlates positively with the attention paid to color words. That is: these heads copy whatever they attend to, and in this setting, they attend to the color words in the test example. Notice that these are the same heads as in IOI (see §C.1). We speculate that these pieces indicates that the model is using distinct subspaces to control what the two types of mover heads are attending to. GPT2-Small does not have this ability; it attends to the IO token using *both* types of

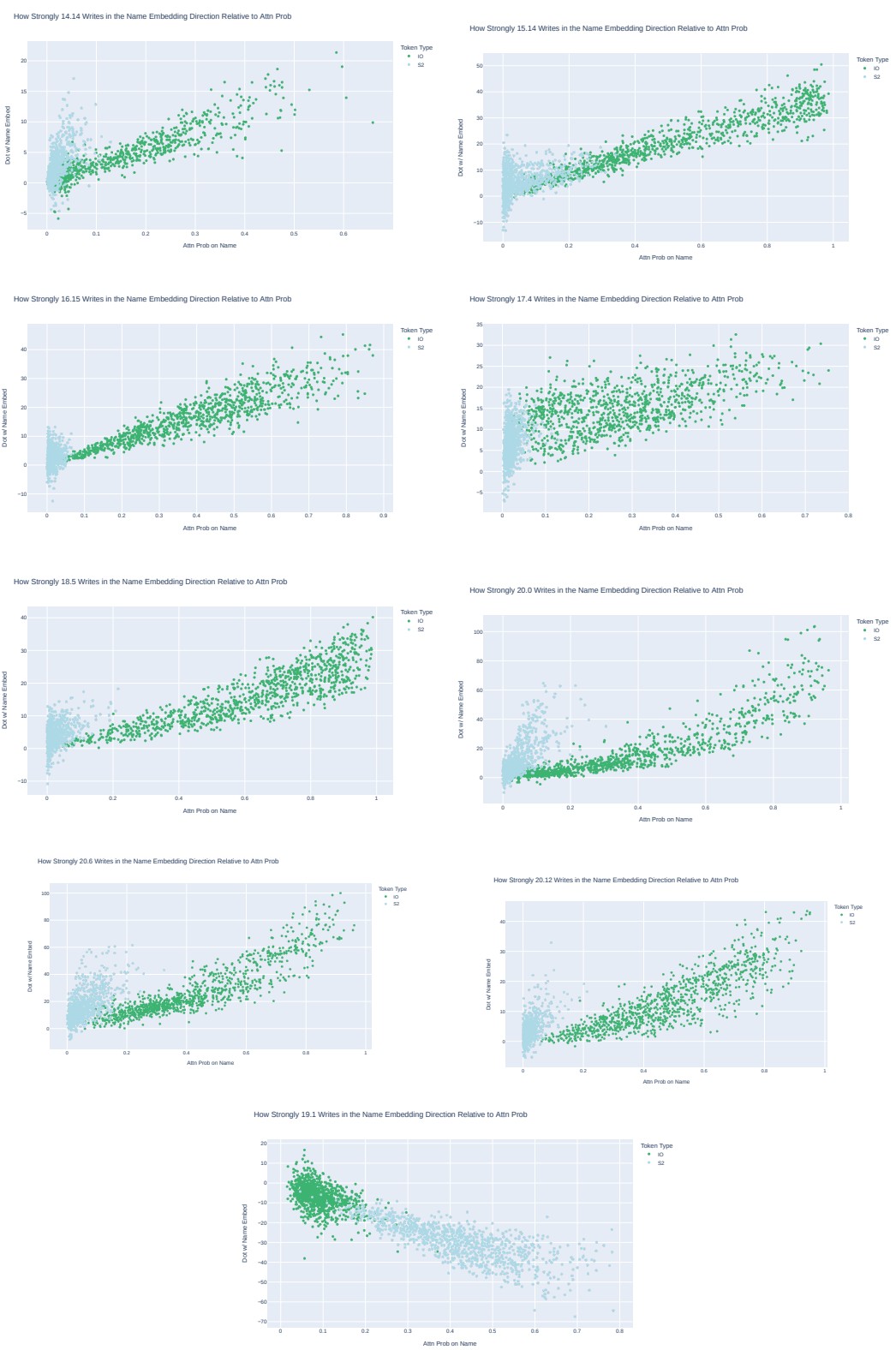

Figure 7: Mover head attention patterns showing copying behavior towards the IO token. The negative mover head 19.1 is shown at the bottom with the opposite trend: attending to the subject token and demoting it. A positive correlation means that the more attention a head pays to a token, the more it promotes it as the next word prediction.

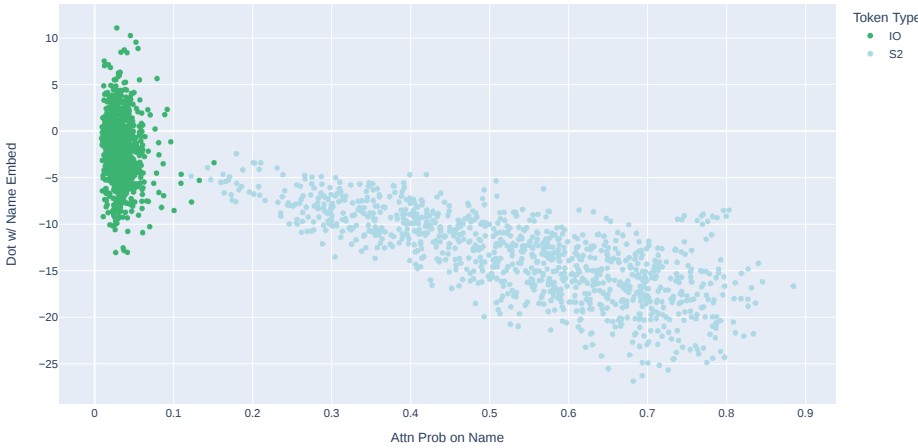

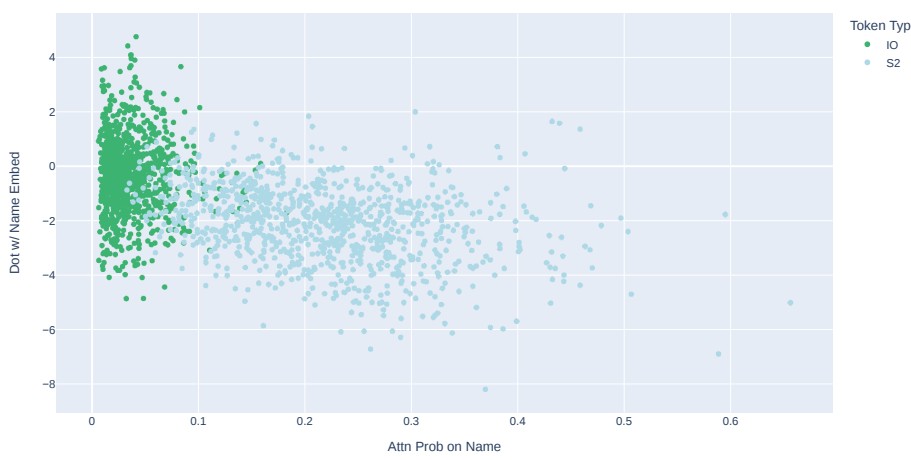

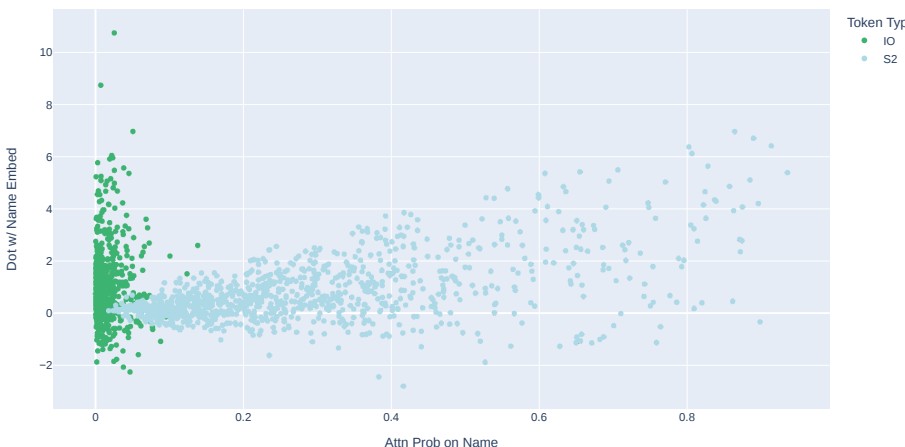

Figure 9: Attention patterns of inhibition heads on the IOI task with respect to the IO and Subject tokens. As in the Colored Objects task, attention head 12.3 writes negatively in the direction of what it attends to, while the others do not have this pattern.

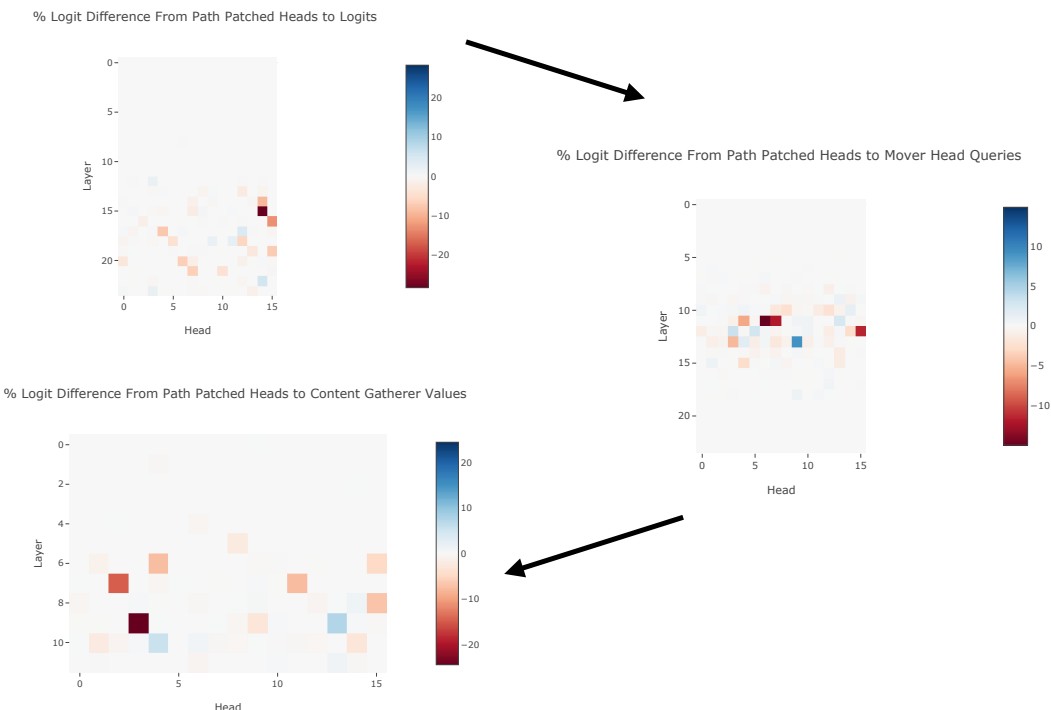

Figure 10: Full path patching results for the Colored Objects task. Color bars show percent logit difference. The lower the number, the more important the head is, as patching it causes the logit difference to drop (i.e., harms performance). We find that the patching process to find the most important attention heads is similar to the one for the IOI circuit and follows the same pattern as described in Figure C. Path patching to the logits highlights the mover heads (15.14 in particular). Path patching to their query vectors highlights the content gatherer heads. Path patching to their values highlights the duplicate token and induction heads most strongly.

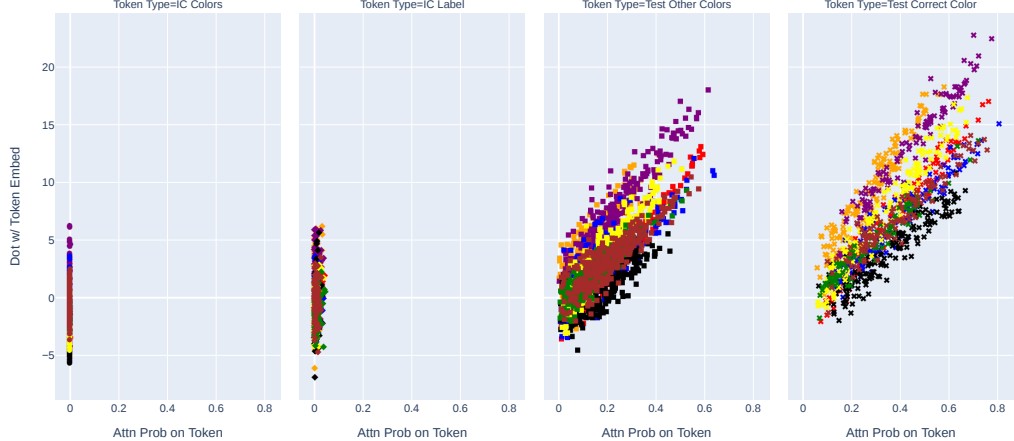

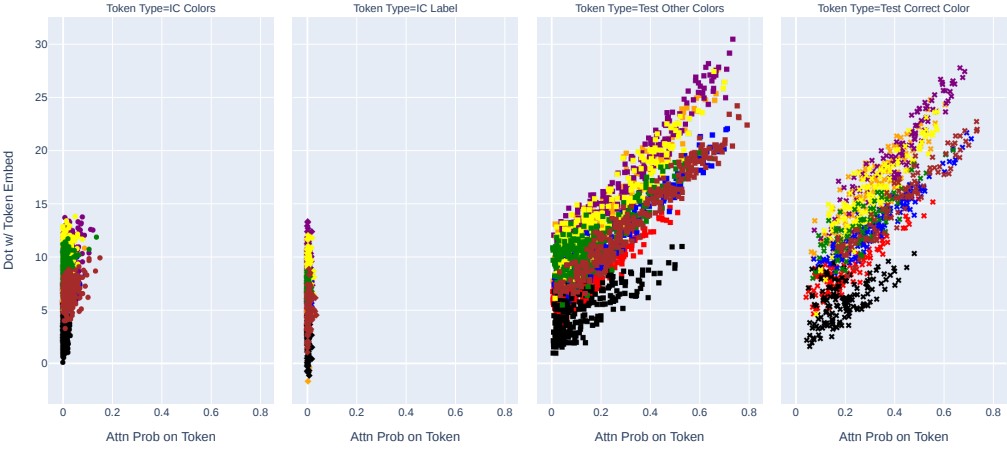

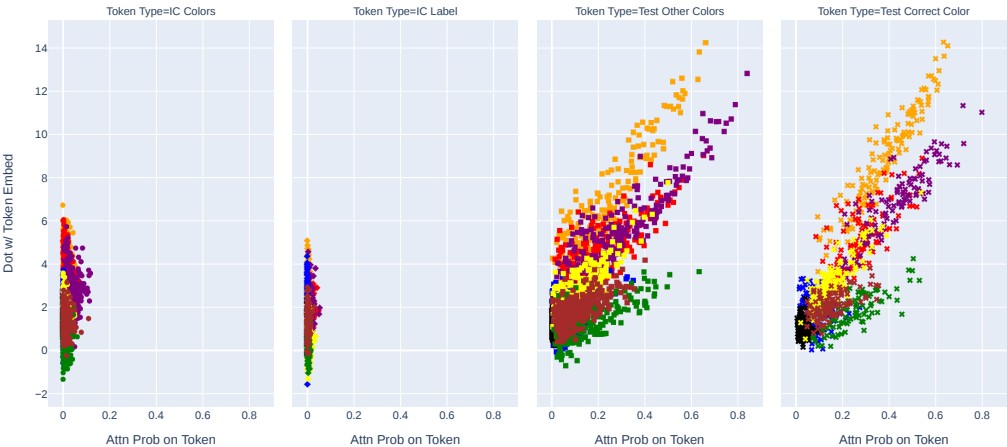

Figure 11: The three most important mover heads in the IOI circuit are also important mover heads in the Colored Objects circuit. In this task they are specifically copying the possible color word options from the test example. Colors of the dots indicate the color of the word being attended to.

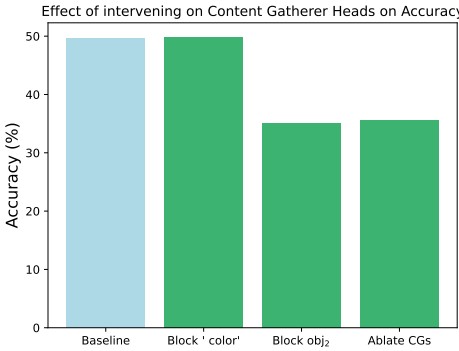

Figure 12: Showing the effect of blocking the attention from the [end] position of the content gatherer heads to various tokens. Blocking attention to the $obj_2$ token causes the model to degrade to about random chance and is similar to zero ablating the content gatherers. Blocking attention to the token " color" does not have this effect.

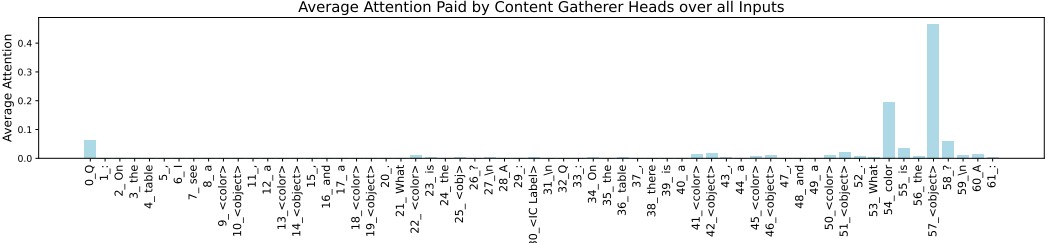

Figure 13: Average attention paid across the entire input sequences by the content gatherer heads from the [end] position. Consistent with our proposed role of these heads, attention is primarily paid to the $obj_2$ position (the object being asked about) and the surrounding context.

mover heads. Thus, it is maybe a result of scale that the larger model is able to separate what these heads attend to. Future work is needed to understand if this is the case, and if so, how the model keeps these bits of information separate.

## D.2   CONTENT GATHERER HEADS

In the main paper, we highlight a sample of the tokens that the content gatherer heads attend to from the last token. The average attention paid to all tokens in the prompts from the [end] position is shown in Figure 2. In Figure 12, we show evidence that the content gatherer heads are accounting for approximately all of the signal telling the model which object to predict the color of. Specifically, blocking attention to the $obj_2$ token causes the model to dip to around random performance between the three color options. Zero-ablating the content gatherer heads both give a similar performance drop. Interestingly, blocking attention to the token " color" does not affect performance negatively at all, but because of the nature of the prompt, there are multiple places that the model can look to gather that the task is to predict a color word, so we can not conclude how important attending to this token actually is for the content gatherer heads.

## D.3   WHAT ARE THE INHIBITION HEADS DOING?

Here we include the attention scatter plots for the three inhibition heads (12.3, 13.4, 13.13) on the Colored Objects task in Figure 14.

## D.4   WHAT IS THE NEGATIVE MOVER HEAD DOING?

We show behavior of the negative mover head on the Colored Objects task in Figure 16.

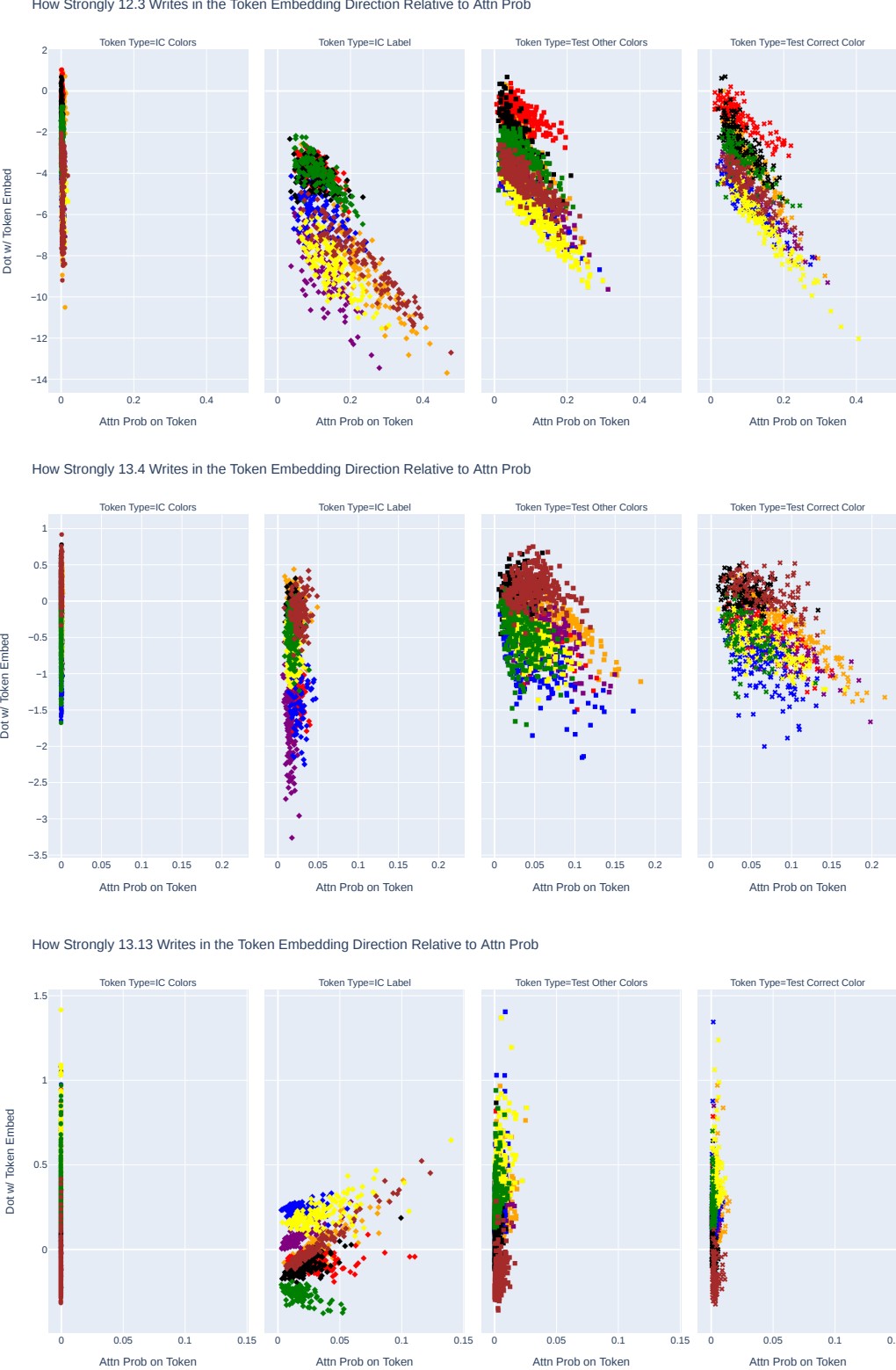

Figure 14: The three inhibition heads exhibit slightly different behaviors, tending to attend towards the previous question's answer (following the "A:") or the color options in the test example. We do not see the model selecting towards inhibiting only the wrong answers, but the attention to the test color tokens is relatively low and split among all of these options. Colors indicate the color of the word being attended to.

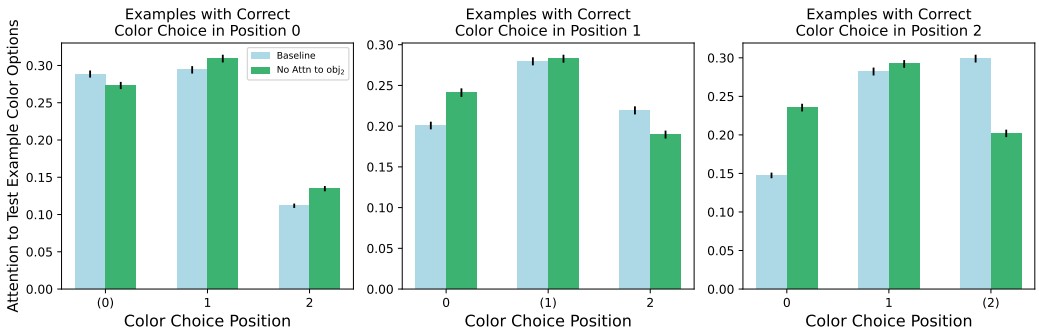

Figure 15: As shown in the main paper, blocking attention in the content gatherer heads from the [end] position to the $obj_2$ token (the object in the question "What color is the $obj_2$?") reduces performance to randomly selecting between the three color options. This figure shows how blocking this attention redistributes the attention paid by the top five mover heads. When the correct answer is listed as the first or third (index 0 and 2) options listed in the prompt, this reduces attention to the correct answer and increases attention to the wrong answers. This is not the case when the correct answer is the middle option. The model tends to be sensitive to the position of the colors presented, which we discuss in Appendix F.

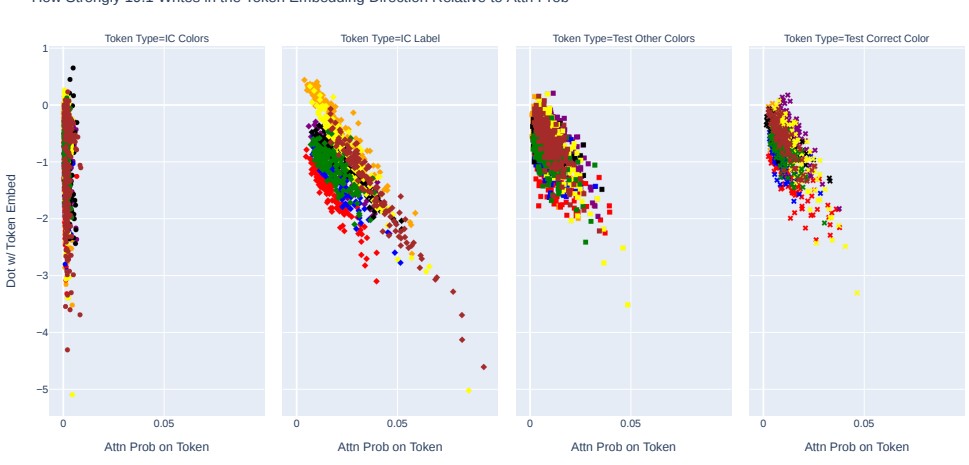

Figure 16: The negative mover head that is highly important in the IOI task has a congruent, but less impactful role in the Colored Objects task. It splits a small amount of attention to the color words and to the answer token of the in-context example. Attending to the last answer would prevent the model from simply repeating the answer to the previous question. The rest of the attention tends to go to the very first token in the prompt.

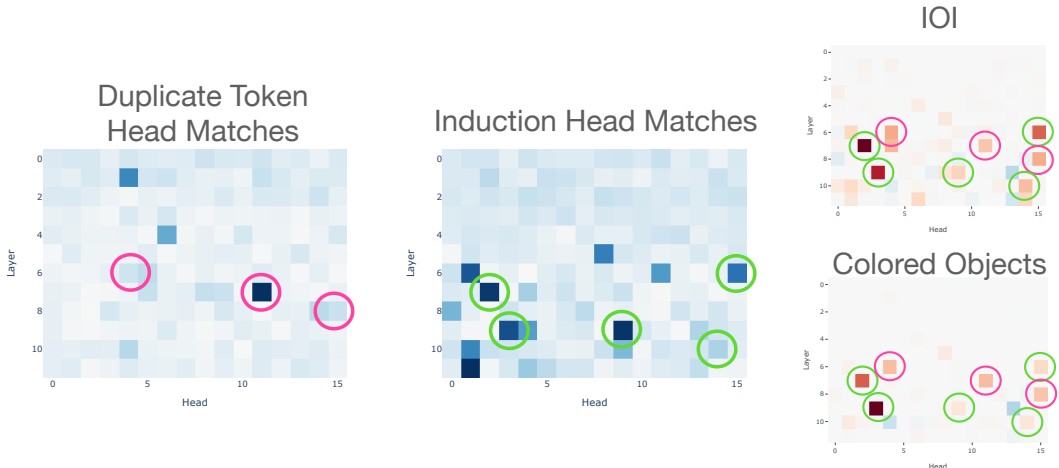

Figure 17: Left two graphs show the induction and duplicate token head matches on random repeating patterns of text, right two show the path patching results for the paths to the inhibition or content gatherer heads, of which the most important heads are strong duplicate token or induction heads.

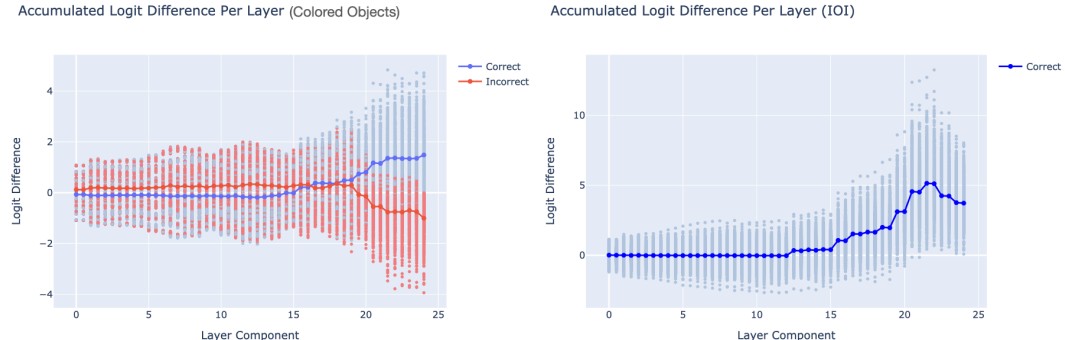

Figure 18: Cumulative logit attribution for both tasks. Here, higher is better: a positive logit difference means the correct answer is more likely than a wrong answer. Measurements for each datapoint are shown as dots, while the average are depicted with lines. For each layer, a measurement of the next token prediction is taken before and after the attention layer. The final measurement is directly before the logits, thus giving us 49 layer components.

## E    DUPLICATE TOKEN AND INDUCTION HEADS

Path patching to the value vectors inhibition heads in IOI or the content gatherer heads in Colored Objects shows that duplicate token and induction heads provide the most signal. Besides just examining the attention patterns, we confirm this quantitatively with the TransformerLens (Nanda & Bloom, 2022) head detector test for duplicate and induction heads using random repeating token sequences as the inputs. In Figure 17, we show that there is a large overlap of important heads at this stage in path patching for both tasks, and duplicate and induction heads.

## F    ADDITIONAL SOURCES OF ERROR IN COLORED OBJECTS

Aside from the circuit-level problems, which prevent GPT2 from accurately inhbiting attention to the incorrect color choices, we find that some circuit components are simply over sensitive to certain features or patterns in the input. As an example, we find that attention head 19.15, which is a copying head, is overly active when the color brown is in the test example (and to a lesser extent, orange), which causes the model to erroneously promote it as the answer. This head even attends to and promotes the token when it appears in the in-context example, which is one of the only examples we

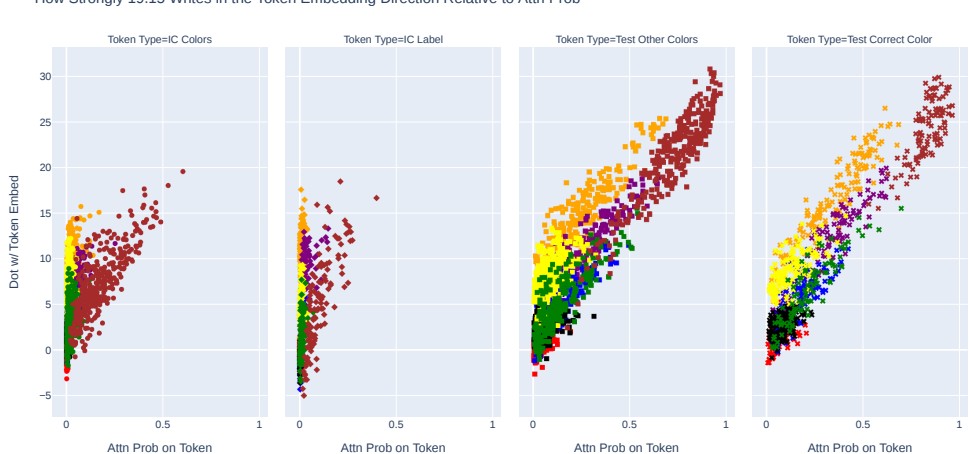

Figure 19: Attention head 19.15 is overly sensitive to attending to and copying the color brown, causing a large portion of errors.

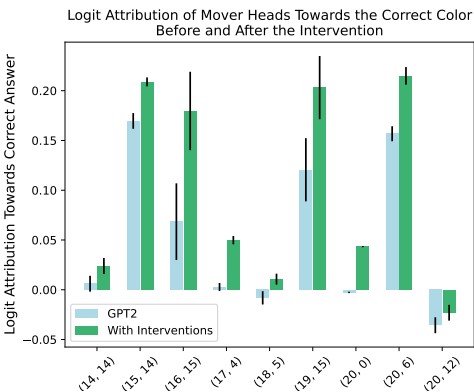

Figure 20: Mover heads contribute more to predicting the right answer after the intervention. Error bars show standard error. While some heads contribute relatively little, we find that heads like 16.15 and 19.15 have a very dramatic increase towards promoting the correct answer.

observe of a head doing something like this. The attention pattern against its output copying score on color tokens is shown in Figure 19.

### F.1 BIAS TOWARDS POSITION IN A SEQUENCE

As is often the case with language models, we observe that GPT2-medium has a bias towards selecting answers based on their position in an input. In the Colored Objects task, the model is sensitive to the order in which the colored objects are presented.

## G    ADDITIONAL DETAILS ON INTERVENTION RESULTS

Here we provide finer-grained analysis of the effects produced by the intervention experiments performed in the main paper. In the main paper, we show the average logit attribution and attention to wrong answers for the mover heads as a result of the intervention. In Figure 20 we show the direct logit attributions of several mover heads on the Colored Objects task. In Figure 22, we show the attention to the correct color answer and the maximally attended to wrong answer for these same mover heads. We can see that the drop in attention to the wrong answer as a result of the inhibition

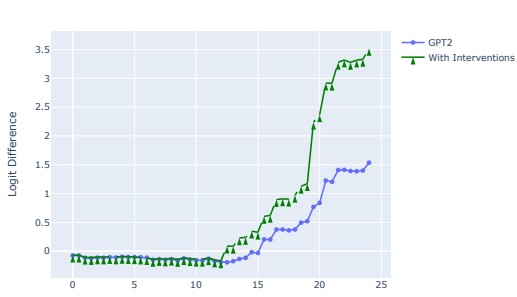

Figure 21: After the intervention, we can see large spikes in logit attribution towards the correct answer as a result of the layer 12 and 19 attention, consistent with the behavior we see in the IOI task. Results are shown for the 496 datapoints that the original model gets correct.

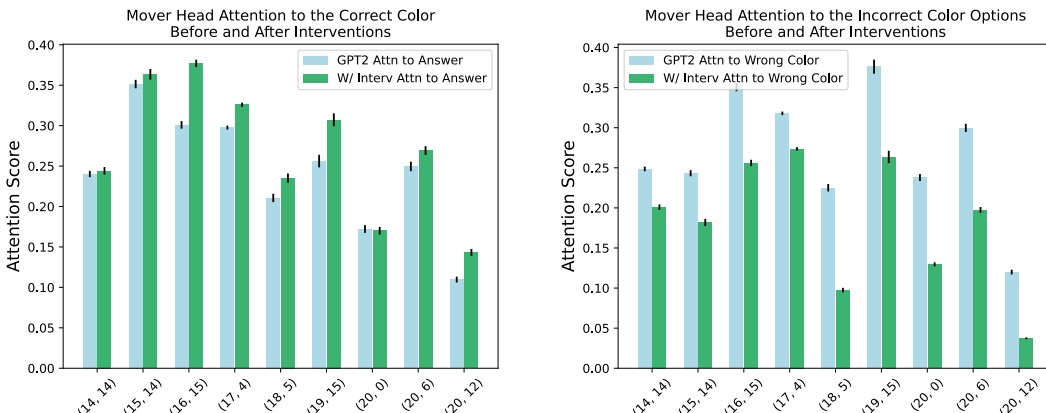

Figure 22: After the intervention on inhibition heads and the negative mover head, the mover heads pay less attention to the incorrect answers and the same or higher attention to the correct answer. This precise behavior is predicted by analysis on the IOI task on GPT2-Small in Wang et al. (2022), but generalizes to our setting, suggesting that the circuit generalizes beyond the IOI task. Error bars show standard error.

head intervention is greater than the increase in attention to the right answer, which is in line with the behavior of the IOI circuit.

We also include the accumulated direct logit attribution following the attention and MLP layer outputs across all layers of the model before and after the intervention (Figure 21). Layer 19 attention in particular increases as a result of the intervention on the negative mover head.

## H    DETAILS ON COLORED OBJECTS DATASET CONSTRUCTION

In the modified Colored Objects task, we give the model three colored objects and ask it to name the color of of one of the objects. The objects span a set of 17 common single-token household objects with each being assigned one of eight possible colors. Any object will only appear once in a given example, with each color only appearing once as well. Each input to the model is a one shot example, meaning we provide one example with the label before providing the additional test example. Repeats in colors and objects can appear across these two examples. The objects are as follows:

Objects: "pencil", "notebook", "pen", "cup", "plate", "jug", "mug", "puzzle", "textbook", "leash", "necklace", "bracelet", "bottle", "ball", "envelope", "lighter", "bowl"

## I  CIRCUIT OVERLAP IN ADDITIONAL TASKS

Comparing the IOI and Colored Objects tasks present an interesting case study that lets us highlight two different tasks with very similar circuits. In this section, we analyze two additional tasks that vary in their similarity to these two tasks to provide additional reference points for quantifying similarity. The two new tasks are the Greater Than dataset (Hanna et al., 2023) and a simple factual recall task in which the model predicts the capital city of a country. While both tasks are fundamentally different from the IOI and Colored Objects tasks, prior work suggests specific hypotheses about the amount overlap we can expect in these circuits.

**Greater Than Dataset**    In this task, originally introduced in Hanna et al. (2023), the model must predict a viable end year for some fictional event. For example: "The war lasted from the year 1732 to the year 17", where any number between 33-99 is possible, but 00-32 is not. The authors analyze the circuit GPT2-Small uses to solve this task, which makes significant use of the MLP layers. Therefore, we expect the circuit GPT2-Medium uses to have very little overlap with the one used for IOI or Colored Objects. We use the code provided by Hanna et al. (2023) to generate a dataset of 200 examples to analyze.

**World Capitals Dataset**    Given a sentence like "The capital of France is" the model must access information that is not provided in context to predict the right city. This makes the task different from either the IOI and Colored Objects tasks. However, Geva et al. (2023) provide an interpretation for how such factual recall happens in LMs (not through circuit analysis). In their analysis, early MLPs enrich the subject token (the "France" token), and later attention heads extract the capital city attribute by attending from "is" to "France". In attribute extraction, some attention heads attend to subject tokens and copy information out of them. If this is the process that is used for this task, our earlier results might predict that the *mover heads* are reused to perform this operation. We perform path patching on a small dataset 200 pairs of different countries to reverse engineer the capital city prediction process. We filter out countries and capitals that are not tokenized as a single token by GPT2-Medium, giving us 47 unique countries to generate the data from.

### I.1  GREATER THAN CIRCUIT

The results of our path patching experiments on this dataset are shown in Figure 23. We path patch using the probability difference metric for all years (see Hanna et al. (2023) for details), which is analogous to the logit difference metric. We first path patch to the logits from the end position and find the MLPs are most important. From there we path patch to last token position to these MLPs, and then backwards through the last token position from there. The circuit can be summarized as follows:

Attention heads 6.1, 5.8, and 6.15, write into 7.2, 9.9, 11.1, and 11.5, 13.12, 14.14, MLPs 14 and 15. In turn these latter components write into MLPs 17-23. To summarize, the earlier induction heads (7.2, 9.9, 11.1, etc. see Figure 17) attend from the last token (e.g., "17") to the last two digits in the start year (e.g., the "32" in "1732"). 14.14 is a significant mover head in the Colored Objects task and has the same pattern. Similarly to Hanna et al. (2023), the MLPs perform the majority of the numerical comparison needed to predict a reasonable year. As expected, this leads to a very low overlap with the two tasks we study in the main paper, in which the MLPs play almost no part.

**Summary of Overlap:**    The Greater Than task represents a task with very low overlap, but there are still components shared with the IOI and Colored Objects circuits. Induction heads 6.15, 7.2, and 9.9 also contribute significantly in IOI and Colored Objects, but this is due to simply matching an induction pattern (Olsson et al., 2022). Mover head 14.14 plays a similar role in Colored Objects. If we just consider the 18 components in the Greater Than circuit, the Colored Objects task has an overlap of 4/18=22.2% and IOI has an overlap of 3/18=16.7%

### I.2  WORLD CAPITALS CIRCUIT:

The path patching results for the World Capitals task are shown in Figure 24. Given prompts of the form "The capital of France is", we first path patch to the logits at the last token position ("is"), and find that the majority of components that write to the logits are mover heads. We path patch to

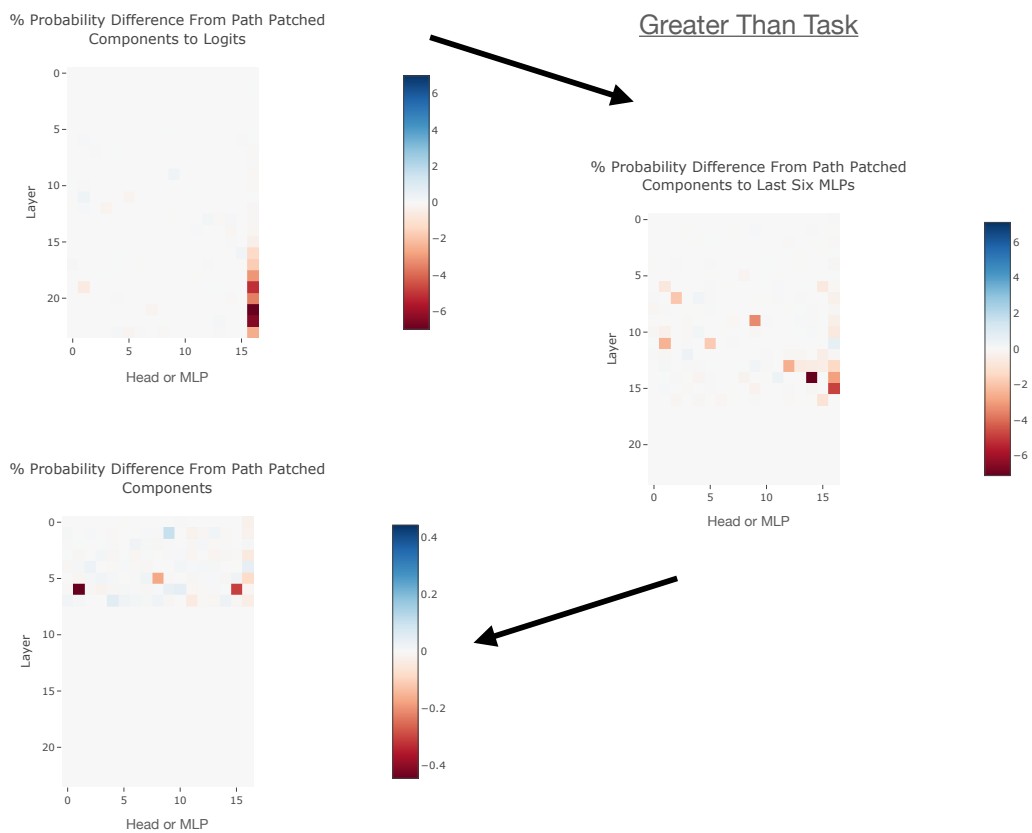

Figure 23: Heatmaps from path patching results from reverse engineering the Greater Than task from (Hanna et al., 2023) on GPT2-Medium. The rightmost column represent the MLPs in each layer.

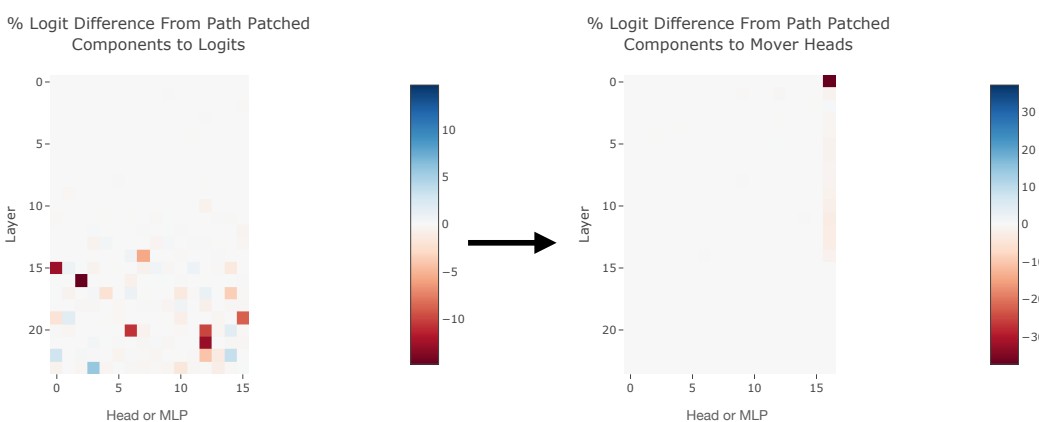

Figure 24: World Capitals path patching results; the rightmost column in the heatmaps represent the MLPs of the layers. Given inputs such as "The capital of France is", the majority of behavior was captured in two steps. First, mover heads attend from the final token position to the country and extract the capital city. The value vectors of the country position for these heads depend on the MLPs in the earlier layers (in particular, MLP 0).

the queries, keys, and values of these heads and find that the values at the country position have the greatest effect on the output of the mover heads, which are predominantly dependent on the MLPs in the early layers. The circuit is summarized below:

MLP 0, 1, 6, and 9-14 build up the representation at the country position of the input. Attention heads 14.7, 15.0, and the mover heads 16.2, 19.15, 20.6, 20.12, and 21.12 attend to the country token and promote the capital city in the logits.

**Summary of Overlap:** Despite the differences between this task and the Colored Objects and IOI tasks, there is significant overlap, particularly with the use of mover heads to copy the final answer. The mover heads that overlap with Colored Objects and IOI are 16.2, 19.15, 20.6, and 20.12. Therefore, 4/7 of the mover heads are shared. Although less prominent, 17.4 and 15.14 also contribute to the world capitals logits which appeared previously as mover heads (Figure 4.1). This aligns with our hypothesis that mover heads would be used for moving information about the capital out of the country. The rest of the circuit relies heavily on the MLPs, so only 4/16=25% of the full circuit components overlap with both the Colored Objects and IOI tasks, demonstrating how different the earlier processing is.

## I.3 CHALLENGES IN QUANTIFYING CIRCUIT OVERLAP

Putting an exact number on how similar two circuits are is a very complex task that we do not attempt to fully answer in this paper. At the moment, the criteria for deciding whether any component is considered in the circuit or not is not very well defined, although there are best practices defined by previous circuit analyses (Wang et al., 2022; Lieberum et al., 2023; Goldowsky-Dill et al., 2023; Hanna et al., 2023) and there is some work exploring automating this process (Conmy et al., 2023). Thus, we do our best to place an accurate number on the overlap as the proportion of shared nodes in the circuit. In the main paper, we study three processing 'steps' in each circuit which also makes the job of assigning similarity easier, but it could be possible that these numbers differ between two tasks for the purpose of other analyses. Our results motivate studying the problem of quantifying circuit overlap in future work.

## J DO OTHER MODELS ALSO USE SIMILAR CIRCUITS FOR THESE TASKS?

We perform a preliminary path analysis of larger versions of GPT2 (Large and XL) by path patching from attention heads to the logits for the IOI and Colored Objects tasks (Figure J). We find that the overlap between the heads used in either task is not as obvious in either model, however it does appear that mover heads are playing a role in both cases. For both models, we compare the top 10 heads in the Colored Objects and IOI tasks. In GPT2-Large, five of these heads overlap, but none overlap in GPT2-XL. These heads are as follows:

**GPT2-Large** (Overlapping heads bolded):

*IOI*: 20.14, 30.13, **27.17**, 29.0, **23.13**, **26.14**, **22.0**, 32.18, 30.10, **21.8**

*Colored Objects*: **22.0**, **23.13**, 23.7, 25.13, **21.8**, 21.14, **26.14**, 24.13, 31.9, **27.17**

**GPT2-XL**:

*IOI*: 41.9, 36.17, 42.7, 28.22, 40.4, 39.9, 39.4, 35.17, 37.15, 38.12

*Colored Objects*: 24.16, 26.20, 28.15, 21.13, 40.18, 29.5, 30.21, 33.18, 31.3, 42.9

It appears that the overlap in the top most important heads decreases as the scale of the model goes up from GPT2-Medium to GPT2-XL, however, we find that the functions of the heads stays roughly the same, meaning that many of these heads are mover heads. We measure this by characterizing the function of each head identified in the IOI circuit on the Colored Objects task. This tells us whether the heads are performing similar roles, even if they don't light up as the most important according to path patching results. In GPT2-Large, the 5 shared heads are all mover heads (21.8, 22.0, 23.13, 26.14, 27.17). In GPT2-XL, even though the exact heads being used are different, 4 of the heads are mover heads (35.17, 36.17, 38.12, 39.4) and 1 is a negative mover head (39.9) on both tasks. As models get larger, there are more 'paths' through the network, which makes it less likely that exactly the same heads are used for different tasks, but it appears there is still overlap in the function of the

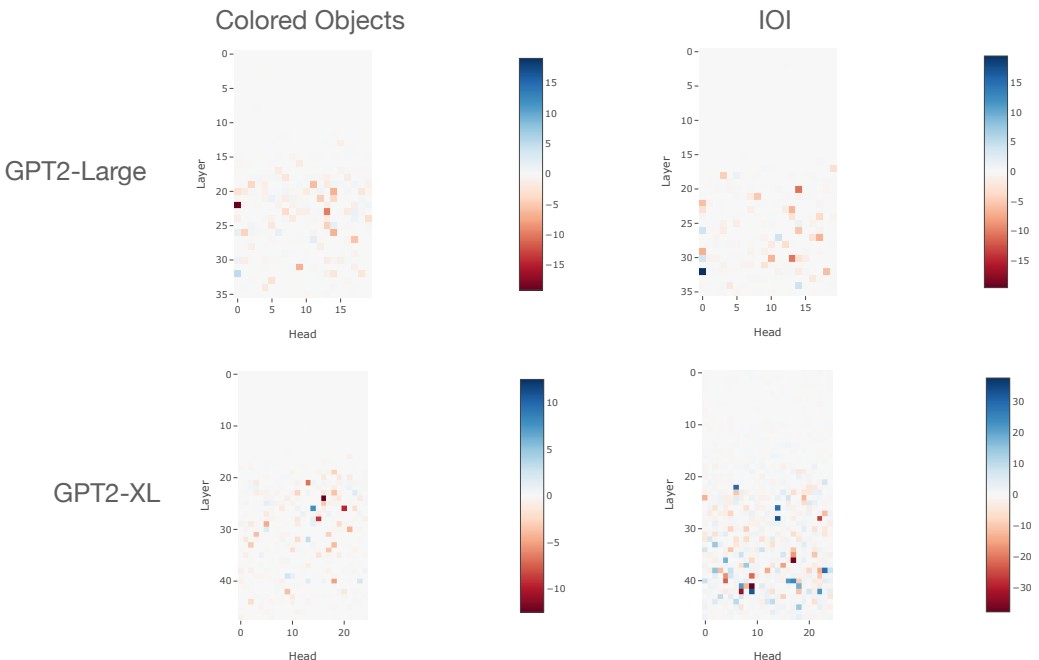

Figure 25: Path patching from attention heads to the logits of GPT2-Large and GPT2-XL. There is some overlap in GPT2-Large, and almost none in the most important heads for GPT2-XL, suggesting that as the number of components increases, overlap is less likely. However, we find that most of the important heads are mover heads in both cases, suggesting the underlying circuits may still be similar functionally even if they take different paths through the models. Colored bars show percent logit difference.

important heads. This is important to keep in mind, as it complicates the picture for analyzing larger models, but the upshot appears that we can still expect some amount of functional overlap between similar tasks.

## K  HOW WAS THE COLORED OBJECTS TASK SELECTED?

The Colored Objects task has no direct connection to the IOI task, so it is not obvious why this was the choice for the point of comparison with the IOI task. This task was being studied for previous work, so it was well understood, and it was noticed that the strategy needed to solve it, i.e., the selection of the answer from a give list of distractors, was qualitatively similar to that used by the circuit described in Wang et al. (2022). We did not path patch the full circuit of any other tasks in the preparation of this study, but preliminary work on other tasks that do not share this connection to the IOI task (e.g., predicting numbers greater than some given integer, predicting the capital of a country) had virtually no overlap with the IOI circuit, indicating that the amount of overlap that we report in the main paper between IOI and Colored Objects is not trivial. However, future work is needed to understand how commonly the mover circuit is used by GPT2, and if the same components behave in different ways under different circumstances that we could not predict here. A limitation of this study is that we are not able to confirm the generality of the circuit beyond the tasks studied here. This type of limitation remains an open problem in the field.

