# OpenReview forum: "Circuit Component Reuse Across Tasks in Transformer Language Models"
_ICLR.cc/2024/Conference — ICLR 2024 spotlight_

### Official Review · Reviewer_4sdK · 2023-10-31

**Soundness:** 3 good
**Presentation:** 2 fair
**Contribution:** 3 good
**Rating:** 8
**Confidence:** 3

**Summary:**

The paper present evidence that there is a substantial circuit reuse between two different probed tasks in GPT2-medium: indirect object identification (IOI) and Colored Objects (CO). Furthermore, it shows a proof of concept intervention to substantially improve the accuracy at a task by "correcting" the behavior of the CO circuit, informed by the behavior of the IOI circuit.

**Strengths:**

- (Significance) The paper addresses an important blindspot of mechanistic interpretability (MI): transferability and generalizability. I believe that the finding of this work will be of substantial interest to the community
- (Clarity) Overall, the paper is structurally well organized and the tasks are clearly presented. However, unfortunately, this does not place the paper in the easy-to-read tier (see below).
- I find the error correction proof of concept particularly interesting, as it shows one possible practical application of MI. To my knowledge, this is a rather novel and worthwhile contribution.

**Weaknesses:**

- (Clarity) My main concern with the paper is the it is not easy to read and follow, especially for people outside the MI field. For once, there is a heavy use of jargon and a liberal use of terms. I'll give some examples below, but I encourage the authors to give a thorough pass of the work and try to "cleanse it" to make it more accessible to a wider audience:
  - `circuit`: although the term is present in large part of the MI literature, I think a refresher and a definition would help
  - `residual stream`/ `write in the residual stream`: likewise, worthwhile defining
  - `[end]` not defined, presume it refers to the last token in the sentence, but unclear
  - `inhibition`: what does it mean specifically?
  -  `noisy signal`: is there a definition for this? does this mean that the head(s) add a random vectors to the residual stream?
  - Labels and titles of Fig 4 are difficult to decipher: what does `Dot w/token embed` mean?
- Some more formalism and equations could help (at least some reader).
     - E.g. one could probably define the  `Dot w/token embed` ..
     - The path patching, even if not a contribution of this work, could be explained more clearly (with a few equations).
- Evidence of the findings (and the storytelling) should find more space in the paper.

Minor
- Suggest to use different typeset for named heads
- Typo: to make to make (pag 7)

**Questions:**

See questions above.
- Do the MLPs play no role in the circuits explained? Are the MLPs even considered in the analysis?

---

> ### Author Response · Authors · 2023-11-20
>
> We would like to thank the reviewer for their comments. We understand that some parts are hard to follow and we would like this work to be accessible to as wide of an audience as possible. It’s true that this type of work makes use of a lot of jargon. We have focused so far primarily on providing additional controls to strengthen the core claims of the paper. We would like to add a glossary in the appendix and properly explain some of the language in the main paper. We definitely appreciate the reviewer’s points on clarity, we will need to take some time to address them all.
>
> >>Do the MLPs play no role in the circuits explained? Are the MLPs even considered in the analysis?
>
> The short answer is no, they pay a negligible role. We can include the actual numbers in the revision, but we wanted to focus on following the setup from Wang, et al. which excludes them.

---

> > ### Comment · Reviewer_4sdK · 2023-11-22
> > **Rebuttal**
> >
> > I thank the authors for their reply and additional experiments; I will keep my score. A glossary and more explanations will help.

---

### Official Review · Reviewer_PvTR · 2023-11-01

**Soundness:** 3 good
**Presentation:** 3 good
**Contribution:** 3 good
**Rating:** 6
**Confidence:** 4

**Summary:**

This paper investigates the similarity of circuits of information flow on two tasks - IOI, and Colored Objects (CO). They study GPT2-Medium, which can do the IOI task successfully, but struggles with the CO task. They reproduce a previous circuit analysis of the IOI task for a different model, as well as for the CO task. They find that many of the same attention heads are important for both tasks and perform the same roles, with a few exceptions. They show if these exceptions are corrected in the CO task to match behavior from the IOI task, the performance can be improved.

**Strengths:**

* The paper has a good understanding of related prior work and presents their work well in context.
* The paper has clear and important motivation - whether circuit analysis and circuit components are transferable across tasks.
* The paper clearly shows there are components that have similar functionality on two different tasks (mover heads, induction heads, and duplicate-token heads).
* The targeted intervention in section 5 is an promising proof-of-concept - showing that understanding the interactions between different components of a circuit may allow correction of undesired behavior when using LLMs for a particular use case. This is the most interesting result of the paper and should be explored more in depth.

**Weaknesses:**

The paper only investigates reuse of components between two tasks, which are pointed out as similar to each other in the paper. The claims in the paper would be strengthened by investigating component reuse across additional tasks that are less similar to the original tasks studied (IOI & Colored Objects). Perhaps studying the behavior of these components on a larger distribution of text (i.e. open web text or some equivalent) that is not task-specific would provide insight into how generally we can understand the inhibition-mover components.

While the intervention in section 5 correctly predicts the interaction between the inhibition heads and mover heads will improve performance, this ignores the other half of the circuit - induction heads, duplicate token heads, and content gatherer heads. A more thorough analysis of the interactions between these elements and the inhibition heads/mover heads would strengthen the claim that these components are part of the circuit. For example, is there an intervention on induction heads/previous token heads (or something else) that would reliably produce the attention pattern you artificially induced into the inhibition heads/negative mover heads? This would show better understanding of the entire circuit, and not just the interaction of the last two sub-components.

The reasons provided for why the inhibition signal “exists but is noisy'' were not very clear/convincing. If the IOI view of inhibition is correct, wouldn't the signal the inhibition heads receive of where to attend (to wrong colors) need to come from the induction heads/previous token heads (or a fuzzy-matching version of the same)?

The methodological contributions of the paper are limited as they mainly rely on existing approaches to discover and validate circuits - path patching, logit attribution, and visualizing attention patterns.

Presentation and Clarity
Minor errors:
Section 2 - Experimental setup mentions “Appendix 4.3” which should be section 4.3, as it is not part of the appendix.
Section 2.1- Colored Objects Task - “other” is repeated twice.
Section 4.5 “deterimine” spelled incorrectly,
Section 7.1 “task” is repeated twice

**Questions:**

Although it is briefly touched upon at the end, there is is an important question to be answered more concretely: why don’t inhibition heads appear to work properly in the Colored Objects setting when they work just fine for the IOI task? One hypothesis might be that the induction heads, which were important for influencing attention of inhibition heads in IOI, aren’t passing forward the proper information but it might be improved with more shots. For example: if you provide some more ICL examples do the inhibition heads start to work as the IOI task would expect them to?

It seems the paper has identified two potential ways through which information about what position to attend can travel to mover heads - through inhibition heads, or through content-gatherer heads.  At least for these two tasks the inhibition-mover subcircuit seems to be more robust than the content-gatherer-mover circuit. This leads to the question: is this always the case for extractive tasks when there are multiple choices that could be copied? Or are there cases where content-gatherer-mover subcircuits work well/better without inhibition heads?

---

> ### Author Response · Authors · 2023-11-20
>
> We’d like to thank the reviewer for the detailed review. We have addressed the reviewers main concerns by including path patching analysis on two new conceptually different tasks and providing more controls to strengthen our main claims. We added a general comment outlining these further.
>
> ## Main Comments
>
> >The paper only investigates reuse of components between two tasks, which are pointed out as similar to each other in the paper. The claims in the paper would be strengthened by investigating component reuse across additional tasks that are less similar to the original tasks studied (IOI & Colored Objects)
>
> To address this, we performed path patching on two additional different datasets (see the new Appendix H). These are the Greater Than task (Hanna, et al. 2023) and a simple factual recall task in which the model must predict the capital of a country (“The capital of France is”). The first represents a task that we expect will have minimal overlap (based on the Hanna, et al. circuit) and the latter tests a specific hypothesis about the reuse of mover heads, as well as provides one additional differing datapoint (see our general comment). In the first task, the overlap is only 16.7% and 22.7% for the IOI and Colored Objects tasks, respectively. The World Capitals task uses 7 mover heads to perform the final copying step, 4 of which appear in the IOI and Colored Objects circuit. The remainder of the circuit uses very different components giving us 25% overlap. We hope this puts the overlap displayed in the main paper into perspective.
>
> > is there an intervention on induction heads/previous token heads (or something else) that would reliably produce the attention pattern you artificially induced into the inhibition heads/negative mover heads?
>
> We agree with the reviewer that this would be satisfying, but this does not appear to work. As reported in Wang, et al. the inhibition heads activate in response to detected duplication of a token (passed by induction/duplicate heads). We show that they work even when no duplication is present, the inhibition heads can still be coerced to work. The action of inhibition heads is fascinating and we are directly following up on them in followup work, but they are complicated and it’s outside the scope of this paper. We believe that our current experiments satisfy our original hypothesis about circuit level overlap, even if it raises some new questions about inhibition heads.
>
> > The reasons provided for why the inhibition signal “exists but is noisy'' were not very clear/convincing. If the IOI view of inhibition is correct, wouldn't the signal the inhibition heads receive of where to attend (to wrong colors) need to come from the induction heads/previous token heads (or a fuzzy-matching version of the same)?
>
> There is a subtle point to correct here: In the IOI view, the induction heads tell the inhibition heads what to write, not where to attend to. Wang et al. show this by path patching to the values of the inhibition heads.
>
> To your final questions: We are glad you also find this interesting, we are hoping to encourage discussion around these topics with this paper. We did try adding more shots, but accuracy does not improve. Our current hypothesis is that since the content gatherer and inhibition heads are close together in the model, the input signal to the inhibition heads is perturbed by the output of the gatherer heads. This gets a bit outside what we aimed to accomplish in this work, but we plan on addressing this.

---

> > ### Comment · Reviewer_PvTR · 2023-11-23
> >
> > Thank you for the additional Appendix H experiments on additional settings; they are helpful.

---

### Official Review · Reviewer_mB6u · 2023-11-07

**Soundness:** 3 good
**Presentation:** 2 fair
**Contribution:** 3 good
**Rating:** 6
**Confidence:** 3

**Summary:**

The paper investigates the generalizability of mechanistic interpretability in language models, focusing on the Indirect Object Identification (IOI) circuit and its application to a different task, Colored Objects. The authors perform an intervention experiment to demonstrate their claims. The work seeks to address the task-specific nature of studies in interpretability and explore the possibility of understanding language models at a higher, more general level.

**Strengths:**

1. **Originality**: The paper addresses an important issue in the field of interpretability by examining the generalizability of mechanistic insights across tasks in language models. This is an original and valuable contribution.

2. **Significance**: Understanding the extent to which language models reuse or learn task-specific circuits has significant implications for the field of machine learning and interpretability. This work provides evidence that the IOI circuits (which is actually more a high-level copying mechanism than something related to indirect object identification) is also used in a least one other task (Colored Object), suggesting the presence of a general and reusable high level mechanism learned by the LM. It is a significant result as it contributes to addressing the challenge of explaining large language models' behavior.

3. **Quality**: The paper employs a range of techniques, including path patching, attention pattern analysis, and logit attribution, to investigate the proposed mechanisms. These methods demonstrate the authors' commitment to a high-quality analysis. Nevertheless, as noted in the Weaknesses Section, the study could be greatly improved by providing control baselines.

**Weaknesses:**

1. **Experimental Demonstration**: The experiments, as currently presented, may not be sufficient to demonstrate the generalizability of basic algorithms in language models. It is only shown for one circuit (IOI) identified with a relatively small LM (GPT2-Medium) in two toy tasks (IOI and Colored Object). It will be essential to explore other types of experiments or analyses to strengthen this argument (out of the scope of this paper). The authors already acknowledge this limitation in the paper. This is a strong limitation. At the same time, this work is a promising first step, that may foster similar work on the topic and improve our knowledge of LMs. (Therefore, this argument alone should not motivate the rejection of the paper.)

2. **Overlap Measurement**: The paper mentions an overlap in in-circuit attention heads but lacks a detailed explanation of how this overlap is computed. Although "putting an exact number on how similar two circuits are is a very complex task that [the authors] do not attempt to fully answer in this paper," the results in the paper heavily rely on this metric. Thus, authors should (a) provide a more detailed description of the overlap metric; (b) quantify the uncertainty of this metric and the results associated to it; and (c) provide clues to better understand its range. As an illustration, authors wrote that 78% of the IOI circuit is used in the Colored Object task. However, without more information, it is difficult to say how significant this overlap is. Providing control baselines, such as comparing the IOI circuit overlap with a set of random tasks, or computing the overlap regarding another circuit (which could be a random circuit), could help to determine if 78% is high or not. Without such information (uncertainty quantification, control baselines, etc.), the significance of the results is hard to interpret. Note that authors briefly mention such an experiment in Appendix I. This should be expanded and included in the main paper.

3. **Interpretation of Intervention Experiment**: In order to prove that some heads are performing the role of gathered heads, the authors conducted an intervention experiment in the model. They did so by blocking attentions to some words in these heads and then measuring the impact on accuracy. They observed that the intervention causes accuracy to decrease and concluded that it successfully demonstrates the role of these heads. However, once again, this experiment lacks a control baseline. Authors should, for instance, reproduce the same intervention on similar heads (but that are not expected to perform the role of CG) and see if the result is different. If the interventions on random heads do not produce the decrease in accuracy observed with candidates’ heads, then it is reasonable to say that these candidates are CGs. Without these control baselines, it is not possible to draw any conclusions from the intervention.

4. **Related Work**: The section on related work is too concise. To provide a broader context, the authors should expand on related work in the field of language model interpretability, specifically those that address the task-specificity of interpretability findings. This will help readers better understand where this work fits in the existing literature and how it contributes to addressing the challenge of task-specificity in interpretability studies.

5. **Lack of Clarity**: The paper exhibits instances of unclear explanations and visualizations, which can hinder readers' understanding of the research. For instance, Section 4.3 is difficult to follow. In addition, there are issues with legends in figures, as terms like "attn prob on token" and "dot w/ token embed" lack sufficient explanation (e.g. in Fig 4). Furthermore, the inclusion of color information in the figures is not adequately commented upon, leaving readers to wonder about its significance. Figures 6 and 7 (Appendix), could benefit from clearer legends and more informative descriptions. Lastly, explanations in the appendix, such as Appendix B, are difficult to follow. While the complexity of mechanistic interpretability tasks may contribute to this challenge, the authors have an opportunity to improve clarity, which would greatly enhance the paper's accessibility.

**Questions:**

# Major questions:
*I'd be willing to consider increasing my rating if these major points are addressed. In particular, addressing the first two points presented below would significantly enhance the paper's soundness.*

1. **Overlap Measurement**: The paper relies heavily on the concept of overlap in in-circuit attention heads, particularly in demonstrating the generalization of mechanisms. While it is acknowledged that putting an exact number on this similarity is complex, can the authors provide a more detailed description of the metric used for measuring overlap? Additionally, could they quantify the uncertainty associated with this metric and provide insights into the expected range of overlap values? To help readers assess the significance of the 78% overlap, would it be possible to provide reference baselines, such as comparing the IOI circuit overlap with a set of random tasks or comparing it with the overlap regarding another circuit, even if it's a randomly generated one?

2. **Interpretation of Intervention Experiment**: The authors conducted an intervention experiment to demonstrate the role of certain attention heads. However, it is crucial to establish the validity of this demonstration by including control baselines. Can the authors consider reproducing the same intervention on similar attention heads that are not expected to perform the role of gathered heads? This approach would help differentiate between candidate gathered heads and other attention heads that might not have the same function. Without these control baselines, it is challenging to draw meaningful conclusions from the intervention experiment.

3. **Related Work**: The related work section, while present, is relatively concise. To provide a more comprehensive context for the readers, can the authors expand on related work in the field of language model interpretability, particularly focusing on those works that address the task-specificity of interpretability findings? This would help clarify where this research fits in the existing literature and how it contributes to addressing the challenge of task-specificity in interpretability studies. Are there any specific prior works that have addressed or attempted to address the same problem or similar questions, and how does this paper compare to them? In particular, how does this work relate to the induction and duplicate heads found in Olsson et al. 2020?

4. **Clarity**: Can the authors address the issues related to clarity, particularly in Section 4.3 and with the legends in the figures? Could they provide clearer explanations for terms and visual elements like "attn prob on token" and "dot w/ token embed" (Fig 4) ? Additionally, what is the significance of the color information in the figures (e.g. Fig 4), and could the authors provide more context or commentary regarding its use in the visualizations? Lastly, are there plans to improve the clarity of explanations in the appendix, given its complexity, to enhance the accessibility of the research for a broader audience? Clarifying these elements would greatly benefit the overall presentation of the work.

# Additional minor questions:

5. Why does the language model fail to correctly utilize the IOI circuit on the Colored Objects task? Is this failure potentially influenced by the number of distractors in the task? If so, it would be informative to explore varying the number of distractors in the task and analyzing the resulting overlap and accuracy to gain deeper insights into the model's behavior in different contexts.

6. What does the term [end] position refer to in the text? It is mentioned but not clearly explained.

7. In Section 4.5, the paper discusses the top 2% of important heads. Could the authors provide a more detailed explanation of how these important heads are defined, particularly in relation to the path patching technique?

8. Appendix J appears to raise questions regarding the generalizability of results from a small model on a toy task to a more complex task with a larger model. The appendix seems to contradict the main paper's claims. Could the authors provide clarification and elaboration on this aspect? Specifically, how do the results from the small model analysis on a toy task relate to accurate predictions on a more complex task with a larger model? This clarification is essential to ensure consistency in the paper's argument.

---

> ### Author Response · Authors · 2023-11-20
> **Part 1/2**
>
> Thank you to the reviewer for the very thorough review and actionable feedback. We have addressed their initial two points by running additional path patching experiments and providing control baselines for intervention experiments to improve our claims. Details are below.
>
> ## Main Comments:
>
> > As an illustration, authors wrote that 78% of the IOI circuit is used in the Colored Object task. However, without more information, it is difficult to say how significant this overlap is… would it be possible to provide reference baselines, such as comparing the IOI circuit overlap with a set of random tasks?
>
> In the new Appendix H, we run path patching experiments on two additional tasks that are significantly different from each other and the original IOI and Colored Objects tasks. These are the Greater Than task (Hanna, et al. 2023) and a simple factual recall task in which the model must predict the capital of a country (“The capital of France is”). The first represents a task that we expect will have minimal overlap (based on the Hanna, et al. circuit) and the latter tests a specific hypothesis about the reuse of mover heads, as well as provides one additional differing datapoint (see our general comment). In the first task, the overlap is only 16.7% and 22.7% for the IOI and Colored Objects tasks, respectively. The World Capitals task uses 7 mover heads to perform the final copying step, 4 of which appear in the IOI and Colored Objects circuit. The remainder of the circuit uses very different components giving us 25% overlap. We hope this puts the overlap displayed in the main paper into perspective.
>
> > the results in the paper heavily rely on this metric
>
> We respectfully disagree, our results heavily rely on the holistic analysis of the circuits and the functionality of all of the model components, not this number. We rely on the number to communicate a simple message that the overlap is high and use our extensive analysis to prove this point.
>
> > In order to prove that some heads are performing the role of gathered heads, the authors conducted an intervention experiment in the model. They did so by blocking attentions to some words in these heads and then measuring the impact on accuracy… this experiment lacks a control baseline.
>
> To address this, we run three new control experiments. There are 3 content gatherer heads, so we repeat the intervention using three randomly sampled heads that aren’t in the circuit from the layers that the 3 real content gatherers come from (without replacement, so 9 different heads were tried). Across these three control baselines the accuracy barely changes: 49.9 +/- 1%. Recall that the original accuracy is 49.7%. This shows quite decisively that the content gatherers have a particularly strong impact in the way we predict. Thank you to the reviewer for pointing this out.
>
> > The section on related work is too concise.
>
> Agreed, unfortunately we cut back to save the much needed space. After revisions we will see if we can reclaim this space, or at least provide a further related work section in the appendix.
>
> > Clarity: Can the authors address the issues related to clarity, particularly in Section 4.3 and with the legends in the figures? Could they provide clearer explanations for terms and visual elements like "attn prob on token" and "dot w/ token embed" (Fig 4) ?
>
> Yes, this kind of work makes use of a lot of jargon. We would like to add a glossary to the appendix on top of providing better explanations in the main paper. Please allow us to address this in our next revision

---

> > ### Author Response · Authors · 2023-11-20
> > **Part 2/2**
> >
> > ## Minor Questions:
> >
> > >Why does the language model fail to correctly utilize the IOI circuit on the Colored Objects task? Is this failure potentially influenced by the number of distractors in the task? If so, it would be informative to explore varying the number of distractors in the task and analyzing the resulting overlap and accuracy to gain deeper insights into the model's behavior in different contexts
> >
> > Ultimately it seems to come down to the inhibition heads. In the original IOI paper the inhibition heads ‘activated’ in response to duplication detection. Our intervention shows that the inhibition heads work even without duplication (the color words are not duplicated in the prompt). It appears that the inhibition heads are more complicated than was initially believed and getting to the bottom of it is out of the scope of this work. Also, we tried additional distractors but the baseline model performance was too low to be useful.
> >
> > >In Section 4.5, the paper discusses the top 2% of important heads. Could the authors provide a more detailed explanation of how these important heads are defined, particularly in relation to the path patching technique?
> >
> > This is based on the heads that cause the highest logit difference when patched. We use the top 2% per path patching step (e.g., the top 2% most important heads that affect the logits, or the 2% most important heads that affect the queries of the mover heads). We attached a number to this to be more objective and concise, but we would prefer the analysis experiments demonstrating the similarity to be the main point of reference for similarity. Typically what counts as in-circuit vs. out-of circuit is up to the discretion of the researchers, we offer this number as at least some way of leaving our judgment out of it, but it isn’t meant to be perfect.
> >
> > >Appendix J appears to raise questions regarding the generalizability of results from a small model on a toy task to a more complex task with a larger model.
> >
> > We may have miscommunicated these results. We have expanded on this appendix significantly to be more clear. The heads that directly affect the logits are predominantly mover heads in both tasks and models, but they don’t necessarily use the same ones. The point of this analysis is to point out that as models grow, there are many different paths through the network that do the same thing, so the literal overlap goes down, but the processing steps still look very similar. This is perhaps an important consideration for working with larger models, that just because the path patching patterns look different doesn’t mean the models are using different processes to solve two different tasks.
> >
> > >What does the term [end] position refer to in the text? It is mentioned but not clearly explained.
> > This is just the last token in the prompt. So the “:” token in Colored Objects. Good catch.
> >
> > We believe we have adequately addressed the major concerns of the reviewer and strengthened the core claims of the paper. We will continue making smaller changes as the discussion period continues. Thank you for the review

---

> > > ### Comment · Reviewer_mB6u · 2023-11-21
> > >
> > > Dear Authors,
> > >
> > > Thank you for your response to my comments and for providing two additional experiments.
> > >
> > > **My initial concerns have been addressed in two ways:**
> > >
> > > 1. Enriching the experiment with two new tasks, namely "greater than" and "knowledge retrieval." This inclusion provides a reference baseline, aiding in better understanding the significance of a 78% overlap. Moreover, it broadens the study's scope, making it more "general" and less task-specific. While I still believe it might be insufficient to draw overarching conclusions about LMs, I acknowledge that this may be beyond the paper's intended scope.
> > > 2. Complementing the gather head intervention experiment with a control baseline by measuring the intervention's effect with random heads. *However, I suggest using more than the current three random samples in the final version of the paper for a more robust analysis.*
> > >
> > > **Consequently, I am revising my rating to "6: marginally above the acceptance threshold." This paper makes a significant contribution to interpretability in language models by examining the generalizability of mechanistic insights across tasks—an original and valuable contribution.**
> > >
> > > However, as mentioned in my initial comment, **the paper lacks clarity**. The structure and jargon used make it challenging for a non-initiated reader to follow. While I acknowledge the complexity of LM interpretability, a work on interpretability should aim to be understandable. Hence, I cannot provide a rating above 6.
> > >
> > > Thank you again for your efforts.

---

### Official Review · Reviewer_F5uQ · 2023-11-07

**Soundness:** 3 good
**Presentation:** 4 excellent
**Contribution:** 3 good
**Rating:** 6
**Confidence:** 3

**Summary:**

This paper falls into the line of work of mechanistic interpretability of neural networks.

This paper compares the Indirect Object Identification (IOI) subnetwork identified in Wang et al. 2022 in GPT-2 small, with a new Colored Objects task subnetwork identified by this paper.

The paper shows that in GPT-2 medium there is a significant overlap between both subnetworks in terms of the attention heads that are activated. Furthermore, the paper conducts an analysis (with ablations and study of the attention probabilities) of the different functions of the heads in the Colored Objects subnetwork and argues that they follow a human-interpretable algorithm. The places where the Colored Objects subnetwork differs from the IOI subnetwork are argued to be the places where the IOI algorithm and the Colored Objects algorithm differ.

Finally, the paper demonstrates that through a handcrafted manipulation of the internal activations, the accuracy of GPT-2 medium on the Colored Objects task can be improved from ~50% to ~100%.

**Strengths:**

The clarity of the presentation is high. The analyses seem to be of high quality. The research field of mechanistic interpretability is important, so the paper is significant insofar as it is a good-quality contribution to this field.

In my opinion the most interesting contribution of this paper is that via mechanistic interpretability analysis, the authors can get a better understanding of the reason that GPT-2 medium fails on the Colored Objects subtask, and intervene on the internal representations at the appropriate heads to get a good output. It would be interesting if this were explored further in settings beyond Colored Objects, and if this could be part of a general framework for improving LLM performance on reasoning tasks.

**Weaknesses:**

The finding that there are circuits that are reused within transformer networks is not new, since this is known e.g., for induction heads. (This is also pointed out by the authors in their related work section.) So I am having trouble wrapping my mind around what the new conceptual contribution of this paper is:

* In terms of techniques, the identification of the IOI subnetwork in GPT-2 medium reproduces an analysis of Wang et al. (2022) using their path patching method. The Colored Objects task network is identified running the same previously-known path patching method. Here the novelty in identifying this network seems to be mainly in identifying what the different heads in this subnetwork do, and writing this as a human-interpretable algorithm. But this style of analysis already appears in the IOI paper.

* Furthermore, the analysis is mentioned in the appendix to not apply to GPT-2 Large & GPT-2 XL, where there is no significant overlap between the IOI and Colored Objects subnetworks. I appreciate the honesty of the authors in reporting this negative finding, but it seems like a major strike against the phenomenon of circuit reuse advocated by this paper. It would be very interesting if the authors could give a more in-depth explanation of why we could expect GPT-2 Large and GPT-2 XL to not have circuit reuse.

With all this said, I certainly do not want to discourage the authors from this line of work, since I believe that it holds a lot of promise.

**Questions:**

1. Could you please clarify what you mean by "normalized by task per each path pathing iteration" in Figure 3?

2. Could you please clarify what you mean by "due to the number of mover heads in the larger models" in Figure 23?

---

> ### Author Response · Authors · 2023-11-20
>
> We’d like to thank the reviewer for their thoughtful review. We have provided extensive experimentation to establish a point of comparison between circuits in two more simple tasks. Details are below as well as in the general comment. We'd also like to reply to some of the reviewer's specific comments below:
>
> ## Main Comments
>
> >So I am having trouble wrapping my mind around what the new conceptual contribution of this paper is
>
> The question we set out to answer with this paper is “Does circuit analysis actually help us understand language models?” The primary criticism of this type of work is that so far it hasn’t been clear whether reverse-engineering LMs at this level of granularity grants us any higher level understanding of what they do. Circuit analyses all look very different, if it turned out that there weren’t patterns across different datasets/behaviors, this type of interpretability work would end up offering very little. Our paper is the first to perform the comparisons needed to answer positively, and show that we can accomplish a surprising amount of control over the LM as a result. The reviewers point out a few areas where we can strengthen this core claim; in our revised draft we added path patching experiments on two new tasks to provide more reference points and provide a more decisive “yes” to our motivating question.
>
> > But this style of analysis already appears in the IOI paper.
>
> We see this is a strength of the paper, not a weakness. The point of our paper is to perfectly replicate the method used in the IOI paper and change one variable at a time: first the model (switching to GPT2-Medium), then the task (switching to the Colored Objects dataset) and analyze how the results change.
>
> To the point about GPT2-Large and XL, as well as the reviewer’s question about Figure 23: We may have miscommunicated the results here a bit, apologies for that. For both tasks and models, many of the heads affecting the logits are mover heads for both tasks. In, e.g., GPT2-Large, 5 of the top 10 heads are shared between tasks and they are mover heads for both. Most of the other heads also appear to be mover heads (or negative movers in the case of IOI), even if they aren’t shared as the most important between the two tasks. To answer the second question, this is what we mean when we say “large number of mover heads”. In these larger models with many more heads, there are more ‘paths’ through the network, so it seems like literal overlap in the exact heads being used is less likely, but the mechanism used to solve the tasks is the same. Should this be considered circuit-level overlap? To us, this is a consideration when working with larger models, but not a counterpoint to our claims. There is no consensus on this, so we respect the reviewer’s opinion if they still feel the same way, but we think this additional information might be helpful to consider. We have expanded on this appendix section to make this point clearer.
>
>
> ## Other Questions:
>
> >Could you please clarify what you mean by "normalized by task per each path pathing iteration" in Figure 3?
>
> Path patching involves an iteration per ‘step’ in the algorithm. For example, on IOI, we path patch to the logits, identify influential attention heads and then path patch to those influential heads,. This represents two steps in the algorithm for which we get logit difference scores for each head in the model. We normalize the logit difference values in each step so that the logit difference scores between each step are comparable. This is purely to help us visualize the importance of each head and doesn’t affect our interpretation of the results. Please let us know if this isn’t clear.

---

> > ### Comment · Reviewer_F5uQ · 2023-11-21
> >
> > > The question we set out to answer with this paper is “Does circuit analysis actually help us understand language models?” The primary criticism of this type of work is that so far it hasn’t been clear whether reverse-engineering LMs at this level of granularity grants us any higher level understanding of what they do. Circuit analyses all look very different, if it turned out that there weren’t patterns across different datasets/behaviors, this type of interpretability work would end up offering very little.
> >
> > I am not sure I buy this. Aren't induction heads already known to appear in multiple LM circuits? And circuit reuse has also been studied outside of LM models (e.g., specialized neurons such as corner detectors in convnets).
> >
> > > We see this is a strength of the paper, not a weakness. The point of our paper is to perfectly replicate the method used in the IOI paper and change one variable at a time: first the model (switching to GPT2-Medium), then the task (switching to the Colored Objects dataset) and analyze how the results change.
> >
> > I think I see what you mean here. It is indeed a contribution to show that the path patching analysis generalizes to other circuits beyond those studied in the IOI paper.
> >
> > > To the point about GPT2-Large and XL, as well as the reviewer’s question about Figure 23: We may have miscommunicated the results here a bit, apologies for that......  In these larger models with many more heads, there are more ‘paths’ through the network, so it seems like literal overlap in the exact heads being used is less likely, but the mechanism used to solve the tasks is the same. Should this be considered circuit-level overlap? .....
> >
> > Thank you for clarifying. From your response it sounds like there is not as much circuit reuse, literally speaking, for these larger models. Is that correct?

---

> > > ### Author Response · Authors · 2023-11-21
> > >
> > > Thank you for the reply
> > >
> > > >I am not sure I buy this. Aren't induction heads already known to appear in multiple LM circuits? And circuit reuse has also been studied outside of LM models (e.g., specialized neurons such as corner detectors in convnets).
> > >
> > > There are two points that we’d like to make here.
> > > 1.) We do not just show that heads are doing something consistently across contexts, but instead that they interact in predictable ways, *and* that lessons from lower level analysis on smaller models and toy tasks can directly guide interpretability efforts on larger models and more complex tasks. For example, building on the insights from Wang et al. allow us to design the inhibition head intervention. All of this shows that circuit analysis builds towards a higher level of understanding of LMs, not just one-off lessons about how a single model solves a single task.
> > > 2.) More specifically with respect to induction heads:. Induction heads will always fire to the same patterns in text: “A B… A__(B)” (perhaps analogous to an edge detector) so we know exactly when and how they will fire, but not how the information they write out is used. In Wang, et al. they are careful not to overclaim what the _Name_ mover and _Subject_ Inhibition heads do because they are dependent on the context in a way that induction heads are not. So it’s not trivial to show that these perform similar operations in new contexts. We appreciate that the reviewer is probably very familiar with the related work and has maybe even observed reuse of these more complex components themselves, but it’s important to establish rigorously in controlled settings if we are to make progress and convince others that this type of analysis work is worthwhile. Within the community, we have fallen into these kinds of traps before. Take for example, the word embedding analogies findings of [Mikolov, et al. 2013](https://aclanthology.org/N13-1090/) which showed that static word embeddings encode semantic relationships like (Russia - Moscow)+France = Paris. This led to people overgeneralizing their belief in what kinds of analogical reasoning was possible with word embeddings. It wasn’t until years later that it was shown that this breaks down for many types of relations ([Gladkova, et. al 2016](https://vecto.space/projects/BATS/)). This is why this type of work is important.
> > >
> > > >Thank you for clarifying. From your response it sounds like there is not as much circuit reuse, literally speaking, for these larger models. Is that correct?
> > >
> > > In a literal sense, yes, path patching tells us that some heads are contributing more to one task than another, but they are all contributing and are generally being used for the same functionality (moving). If a model has 100 mover heads, is every task that uses them going to assign importance the same way across them all? This was the sort of question that motivated running the experiment. The results show potential pitfalls of using a simple overlap metric and the importance of properly understanding what the heads are actually doing to establish whether functionally similar things are happening. We hope this point is now clear, because we don’t want to mislead the readers into an oversimplified picture of what the models are doing.
> > >
> > > We hope that we can convince the reviewer that this paper is of significant importance to the field. In any case, we appreciate the insightful discussion this has generated. We want this work to encourage the community to engage more with these questions. Further feedback on these points is welcome and as we continue to work on the presentation of our findings, we will clarify some of the points brought so far for the final revision.

---

> > > > ### Comment · Reviewer_F5uQ · 2023-11-21
> > > >
> > > > Thanks for your clarifying response. I definitely think this line of inquiry is valuable, so will raise my score to 6.

---

### Author Response · Authors · 2023-11-21
**Summary of Contributions and New Experiments**

We would like to restate the contributions of this paper and summarize the revisions we made based on reviewer feedback.

# Contributions:
In the general landscape of circuit analysis, there is an emphasis on explaining the widest breadth of model behaviors and tasks. While these lead to interesting insights into how LMs work, the discovered circuits often look and behave very differently. This is concerning because if every task is handled idiomatically, analyzing circuits will not contribute to a higher level of understanding of neural networks. We perform the first controlled comparison to evaluate whether circuit components are reused across tasks, finding that they seem to play high level and interpretable roles across different tasks, suggesting avenues towards understanding the behaviors of LMs.

We reproduce the IOI circuit results from Wang, et al. and extend them to a larger model, and use this to experimentally validate that the circuit used by the Colored Objects task is solved with a highly similar circuit, and that specific components are reused. We use insights from mechanistic interpretability to show that we can predictably control the output of GPT2-Medium to solve the Colored Objects task using manually designed interventions to attention heads.

Our analysis reveals novel insights that extend the understanding of mover heads, inhibition heads, and negative mover heads and provides a control for how these components change with an increase in model size from previous work.

# New Experiments/Results

The reviewers gave very clear feedback that this paper would be strengthened if we included analysis on additional tasks to provide reference points for how significant the overlap between IOI and Colored Objects is. To address this, we added path patching analysis for two new tasks in Appendix H. We believe this addresses the main concerns and greatly strengthens our original claims. We also provide additional controls for the intervention experiments.

## Additional Path Patching Experiments

In order to provide points of reference for how high the overlap between Colored Objects and IOI tasks (C+I) we find is we path patch the circuits used for two simple tasks, the Greater Than (GT) task (“The war lasted from the year 1732 to the year 17” [Hanna, et al. 2023](https://arxiv.org/abs/2305.00586))  and a capital cities prediction task (“The capital of France is”). The GT task represents a task that likely has a very small amount of overlap with the C+I tasks due to the results of Hanna, et al. We choose the world capitals task because it is also a highly different task than (C+I). [Geva, et al. 2023](https://arxiv.org/abs/2304.14767) also characterize a mechanism LMs use to solve tasks like these. In their study, they describe heads that attend to tokens and copy attributes out of them. The described mechanism sounds a lot like the function of mover heads. We thus expect a high amount of overlap in the mover heads but very little in the remaining circuit.

### Results
The overlap between these circuits matched our hypotheses. In the GT task, we saw a reliance on the MLP layers with some overlap in the induction heads. The IOI task and GT have an overlap of 16.7%. In the world capitals task, the model uses mover heads to copy the capital city out of the country token as expected: over half (4/7) of the mover heads are used in the IOI task, providing another example of component reuse that was not originally in the paper. The remainder of  the circuit uses MLPs to enrich the country token. The the final overlap is 25% with IOI.

To summarize the overlap statistics with IOI, we have:
Colored Objects: 78%
World Capitals: 25%
Greater than: 16%

## Additional Controls for Intervention Experiments
One reviewer pointed out that we did not convincingly show the special role of the content gatherer heads. In our intervention, we block attention to the object being asked about in the Colored Objects prompts from the content gatherer heads and show that accuracy drops to near random performance, but this does not preclude the possibility that simply intervening on the model causes a sharp decrease in performance. To address this, we run three new control experiments. There are 3 content gatherer heads, so we repeat the intervention using three randomly sampled heads that aren’t in the circuit from the layers that the 3 real content gatherers come from (without replacement, so 9 different heads were tried). Across these three control baselines the accuracy barely changes: 49.9 +/- 1%. Recall that the original accuracy is 49.7%. This shows quite decisively that the content gatherers have a particularly strong and causal impact on the predicted token in accordance with our description.




Lastly, we would like to thank all of the reviewers for their effort and high quality reviews

---

### Meta-Review · Area_Chair_ajJS · 2023-12-03

**Metareview:**

This paper follows up on Wang 2022's discovery of a specific circuit, or sub-graph of nodes, that performs Indirect Object Identification. The current paper shows that the same circuit also appears in a larger model, and, importantly, that it is involved in solving a distinct but functionally related task, namely Colored Objects.

While in a sense incremental with respect to Wang 2022's work, the paper is exciting as it provides the first (to the best of my knowledge) clear evidence of circuit reuse for non-trivially related tasks in a pre-trained mode. It thuys makes a strong contribution to the current, slow and painful progress of mechanistic interpretation of language models.

There is some concern about the generality of the results, and the fact that the network behaviour seems to change with scale, and I hope the authors will acknowledge such issues more clearly in the final revision. Still, I think this is a helpful contribution that should appear at ICLR.

**Justification For Why Not Higher Score:**

While the result presented in the paper is exciting, it is somewhat incremental, and it might only appeal to a subset of the ICLR audience.

**Justification For Why Not Lower Score:**

For those who care about mechanistic interpretability (and even curious bystanders), this is a cool result, and people should know about it!

---

### Decision · Program_Chairs · 2024-01-16

Accept (spotlight)